



# Measuring dry deposition of ammonia using flux-gradient and eddy covariance methods with two novel open-path instruments

Daan Swart[1], Jun Zhang[2], Shelley van der Graaf[1], Susanna Rutledge-Jonker[1*], Arjan Hensen[2], Stijn Berkhout[1], Pascal Wintjen[2,4], René van der Hoff[1], Marty Haaima[1], Arnoud Frumau[2], Pim van den Bulk[2], Ruben Schulte[1,3], Thomas van Goethem[1]

[1] National Institute for Public Health and the Environment (RIVM), P.O. Box 1, 3720 BA, Bilthoven, the Netherlands
[2] Netherlands Organisation for Applied Scientific Research (TNO), P.O. Box 15, 1755 ZG, Petten, the Netherlands
[3] Wageningen University & Research (WUR), P.O. Box 47, 6700 AA, Wageningen, the Netherlands
[4] Thünen Institute of Climate-Smart Agriculture, Bundesallee 68, 38116 Braunschweig, Germany

*Correspondence to: susanna.jonker@rivm.nl

**Abstract.** Dry deposition of ammonia ($NH_3$) is the largest contributor to the nitrogen deposition from the atmosphere to soil and vegetation in the Netherlands, causing eutrophication and loss of biodiversity. Yet, data sets of $NH_3$ fluxes are sparse and in general have monthly resolution at best. An important reason for this is that measurement of the $NH_3$ flux under dry conditions is notoriously difficult. There is no technique that can be considered the golden standard for these measurements, which complicates testing of new techniques and judging their quality.

Here, we present the results of an intercomparison of two novel measurement setups aimed at measuring dry deposition of $NH_3$ at half-hourly resolution. In a five-week comparison period, we operated two optical open-path techniques side by side at the Ruisdael station in Cabauw, the Netherlands: the novel RIVM-miniDOAS 2.2D using the aerodynamic gradient technique, and the novel commercial Healthy Photon HT8700E using the eddy covariance technique. Both are open-path optical instruments, leaving $NH_3$ in the air during measurement. Otherwise, they are widely different in their measurement principle and approach to derive deposition values from measured concentrations. The two different techniques showed very similar
results when the upwind terrain was both homogeneous and free of nearby obstacles ($r = 0.87$). The observed fluxes varied from a deposition of ~80 ng $NH_3$ m$^{-2}$ s$^{-1}$ to an emission of ~140 ng $NH_3$ m$^{-2}$ s$^{-1}$. We obtained similar results from two widely different techniques, both in absolute flux values as in their temporal pattern, which substantiated that both instruments were able to measure $NH_3$ fluxes at high temporal resolution for a consecutive period of at least several weeks. However, for wind
directions with nearby obstacles, the correlations between the two techniques were weaker. Moreover, the technical performance (e.g., uptime, precision) and practical limitations of both systems were discussed. The uptime of the miniDOAS system reached 100% once operational, but regular intercalibration of the two instruments was applied in this campaign (35% of the 7-week uptime). Conversely, the HT8700E did not measure during, and shortly after, rain, and the coating of its mirrors tended to degrade (21% data loss during the 5-week uptime). In addition, the HT8700E measured $NH_3$ concentrations proved
sensitive to air temperature, causing substantial differences (range: -15 to + 6 μg m$^{-3}$) between the two systems.

To conclude, the miniDOAS system appeared ready for long-term hands-off monitoring. The current HT8700 system, on the other hand, had a limited stand-alone operational time under the prevailing weather conditions. However, under the right circumstances, the system can provide sound results, opening good prospects for future versions, also for monitoring applications. The new high temporal resolution data from these instruments can facilitate the study of processes behind $NH_3$
dry deposition, allowing improved understanding of these processes and better parametrization in chemical transport models.



## 1 Introduction

Human alteration of the global nitrogen cycle through agricultural, industrial, and combustion processes has led to unprecedented levels of reactive nitrogen ($N_r$) in the Earth system (Galloway et al., 2021; Fowler et al., 2013). Besides benefits like increased food production, losses of $N_r$ have a range of detrimental effects on both the environment and human health
(Sutton et al., 2011; Erisman et al., 2015). For example, many areas worldwide now experience increased levels of deposition of reduced nitrogen in the form of ammonia ($NH_3$) and ammonium ($NH_4^+$), which contribute to acidification and eutrophication of both terrestrial and aquatic ecosystems leading to the loss of biodiversity and ecosystem services (Bleeker et al., 2011; Bobbink et al., 2010). In addition, atmospheric $NH_3$, when transformed into $NH_4^+$, can form particulate matter in the atmosphere, which is increasingly recognised as a threat to human health (Giannakis et al., 2019; Lelieveld et al., 2015).

Gaseous $NH_3$ can be emitted from and deposited onto the Earth's surface: the exchange is bi-directional. $NH_3$ volatilisation during agricultural activities like manure and slurry storage and application, plus emissions from farm buildings, represent by far the largest source of anthropogenic $NH_3$ emissions (Reis et al., 2009; Sutton et al., 2013; Damme et al., 2014). With regards to deposition, dry deposition of $NH_3$ is an important component. In the Netherlands for example, it typically accounts for more than a third of the total $N_r$ deposition (Hoogerbrugge et al., 2020). The remainder is made up of dry deposition of oxidised
forms of nitrogen, and wet deposition. Accurate quantification of biosphere-atmosphere exchange of $NH_3$ is therefore essential to increase our understanding of $NH_3$ budgets at regional and global scales, to study relevant processes at high time resolution, monitor trends, measure the effectiveness of mitigation efforts, and improve and validate air quality and deposition models.

Despite the relevance of high-quality measurements of $NH_3$ exchange, relatively few direct long-term continuous measurements have been reported. Dry deposition of $NH_3$ can be highly variable in time and space and depends on a variety
of site-specific parameters like canopy wetness, leaf area, and surface roughness (Flechard et al., 2011). Micro-meteorological techniques provide the most direct estimates of dry deposition, but these measurements each present their technical challenges and generally require substantial expense and labour. The most commonly used micrometeorological methods include the aerodynamic gradient method, modified Bowen ratio method, relaxed eddy accumulation, path-integrated techniques coupled with backward dispersion models, and eddy-covariance (see overview in e.g. Trebs et al., 2021; Fowler et al., 2009; Hensen,
65 2011).

The aerodynamic flux gradient method (AGM, also 'profile method') has delivered the majority of the $NH_3$ dry deposition data worldwide. Most of these measurements were done using wet chemical instrumentation. Wyers et al. (1993) describe the wet chemical denuder system that was used later in the Netherlands above forests (Erisman and Wyers, 1993) and over heathland (Nemitz et al., 2004) and in European intercomparison campaigns like the Braunschweig campaign (Sutton et al.,
2009). The AGM method is still commonly used for inferring half-hourly $NH_3$ fluxes, now also using optical $NH_3$ measurement systems (e.g. Kamp et al., 2020; Loubet et al., 2012; Wolff et al., 2010b; Walker et al., 2020; Hassouna et al., 2016; Ramsay et al., 2018). In the AGM method, surface-atmosphere exchange fluxes are derived from measurements of vertical concentration differences ($d_{NH_3}$) combined with a measure of vertical turbulent transport (Loubet and Personne, 2016; Prueger and Kustas, 2005). Drawbacks of AGM (listed by Trebs et al., 2021; Loubet et al., 2013) include potentially biased gradients
under non-stationary conditions if sequential sampling at multiple heights using one monitor is required (Kamp et al., 2020), or, if using multiple monitors, the need for regular side-by-side comparisons to accurately determine and correct for any potential systematic difference (bias) between monitors (Wolff et al., 2010a; Walker et al., 2013). Because of its reactivity and solubility in water, $NH_3$ is sticking to inlet walls, filters and measurement cells (e.g. Von Bobrutzki et al., 2010; Norman et al., 2009; Twigg et al., 2022). Systems with an inlet system, either wet chemical or optical with a closed sampling cell, will be
prone to sampling carry-over $NH_3$ due to $NH_3$ being adsorbed and released again inside the measurement system. This is affecting the measured concentration gradient and thus the fluxes derived with the AGM technique. Finally, a drawback of AGM is the need to rely on empirical stability corrections, which are based on relationships found for sensible heat, but assumed to be the same for trace compounds like $NH_3$.





Open-path (OP) analysers have no sampling tubes and provide a way of measuring concentration in situ, without interfering
with the airflow. A long-line averaging open-path gas analyser allows measurements of path integrated $NH_3$ concentrations at
a high time resolution. Optical analysers now available include those based on Fourier Transform Infrared (FTIR) (Sintermann
et al., 2011; Flesch et al., 2016), tuneable diode laser TDL (Bai et al., 2022) or differential optical absorption DOAS
spectroscopy (Volten et al., 2012b; Sintermann et al., 2016). These instruments can be used to measure the difference in $NH_3$
concentration between two vertically offset paths, either in slant configuration (e.g. Bai et al., 2021; Flesch et al., 2016) or in
two parallel horizontal paths. In the Netherlands, several experiments have taken place using two DOAS systems to measure
$d_{NH_3}$ (Wichink Kruit et al., 2010; Volten et al., 2012a; Schulte et al., 2020). Over the last year, the more recently developed
miniDOAS (Berkhout et al., 2017) has been adapted and improved to meet the high sensitivity required for flux gradient
measurements of $NH_3$ (Wolff et al., 2010a; Foken, 2017).

Eddy covariance (EC) is the preferable technique for measuring the surface-atmosphere gas exchange of any compound
because it provides the most direct measurement. For many greenhouse gases, it has already become the reference method
(Mauder et al., 2021; Baldocchi et al., 2001). However, EC requires fast (<0.1 s) and precise concentration measurements,
which is particularly challenging for $NH_3$. In recent years, several studies have reported measurements of the $NH_3$ flux using
closed-path (CP) analysers based on different techniques like a tandem mass spectrometer (Shaw et al., 1998), high-
temperature chemical ionization mass spectrometry (Sintermann et al., 2010), and laser-based solutions like the tuneable
infrared diode laser differential absorption spectroscopy (TILDAS) technique (Brodeur et al., 2009; Famulari et al., 2004;
Whitehead et al., 2008) or a quantum cascade laser (QCL) instrument (Ferrara et al., 2016; Zöll et al., 2016; Moravek et al.,
2019; Whitehead et al., 2008). However, the 'stickiness' mentioned above presents challenges there, because the use of inlet
tubing leads to loss of fast variations in the signal. The amount of damping of the signal is highly variable, and depends on
temperature, humidity, and cleanliness of the tubing, among other things. Correcting for this high-frequency loss introduces
additional uncertainty (Moravek et al., 2019).
So far, two folded-path open-path instruments have been reported in the literature for eddy covariance measurements of $NH_3$.
Besides the benefit compared to the CP setup of not needing an inlet tube, such systems generally have much lower power
requirements and the less bulky installation may allow for a more portable and adaptable setup also at more remote sites.
The first OP EC $NH_3$ analyser is the QCL-based instrument developed by Princeton University, and improved from the original
design presented in Miller et al. (2014) over various deployments (Sun et al., 2015; Pan et al., 2021). More recently, a similar
instrument has become available from Healthy Photon Co. Ltd., Ningbo, China: model HT8700 (Wang et al., 2021).
Limitations of open-path EC flux measurements include interference from contamination by dust and rainfall, and degradation
of mirrors over time (Wang et al., 2021). Since this technique is evaluating the net flux by measuring concentration levels in
both up and down going air that passes the sensing volume both in small, high frequency (> 5 Hz) eddies and in slow (> 10
minute) large turbulent eddies the method needs corrections for differences in air density between up and down going air linked
with heat and humidity transport. Similar to closed-path EC gas analysers, not all sizes and frequencies of eddies are measured
completely and therefore (high and low frequency) spectral corrections are needed.
Both micrometeorological methods (AGM and EC) share additional limitations to those mentioned above, such as the need
for a homogeneous upwind fetch to avoid local advection errors. They also require steady-state conditions, well-developed
turbulence, with no change in vertical flux with height (Loubet et al., 2013; Mauder et al., 2021).
In this study, we measured bi-directional $NH_3$ fluxes in a field campaign of seven weeks from August 24[th] to October 11[th],
2021 over grassland at the Cabauw research site in the Netherlands, during which both deposition and emission events were
encountered. During a period of 5-weeks (August 27[th] to October 1[st]), we compared measurements of $NH_3$ concentrations and
fluxes from two open-path instruments: the RIVM miniDOAS 2.2D using the AGM, and the commercial HT8700E from
Healthy Photon Inc. using the EC technique. This was the first time either one of these systems was compared to another setup.



Here, we describe the uptime and performance of both setups and compared the results of both concentration and flux measurements of $NH_3$. Moreover, potential sources of errors, challenges encountered and the current suitability and future potential of the different setups for long-term in-situ measurement under field conditions are discussed.

## 2    Campaign setup and site

### 2.1    Site description

$NH_3$ measurements were performed at the Cabauw site for atmospheric research (51.97034° N, 4.92559° E, elevation –0.7 m a.s.l.). The site is operated by Royal Netherlands Meteorological Institute (KNMI) and has been an atmospheric research station for over half a century (Bosveld et al., 2020). It hosts an extensive suite of meteorological and atmospheric instrumentation,

some on the 213 m high mast on the facility. It is also one of the stations of the Dutch National Air Quality Monitoring Network and since 2019 part of the Ruisdael observatory (https://ruisdael-observatory.nl/the-rita-2021-campaign/, last access date: 20 Apr 2022). The site is 15 and 25 km away from the urban areas of Utrecht and Rotterdam, respectively (Figure S1). The area is completely flat (slopes less than 3%), with ribbon-shaped villages built along minor watercourses. Land use in the general area is predominantly agricultural, with most plots intensively managed grassland with an average vegetation height of 0.1 m

used to graze cattle or sheep, or for silage. The soil consists of 35-50% river clay in the top 0.6 m, overlying a thick layer of peat (Bosveld, 2020). The soil of the top layer (0–0.15 m) has a bulk density of 1.14 g cm$^{-3}$ (Jager et al., 1976). The measurement site is drained by narrow (1-3 m) parallel ditches, which are on average 40 m apart. The site policy is to keep the grass short by having it grazed by sheep. To prevent sheep from blocking the miniDOAS optical paths or from damaging instrument cables, the measurement area was secured with a low profile electric fence. During the campaign, sheep were

grazing the plots of land immediately surrounding the measurement site. These sheep often grazed within 100 m north to northeast of the instruments with about 50 animals per hectare. Furthermore, the plots surrounding the research site were occasionally manured by local farmers, which was allowed up until September 15 (RVO, https://www.rvo.nl/onderwerpen/mest/gebruiken-en-uitrijden/wanneer-uitrijden, last access 12 May 2022).





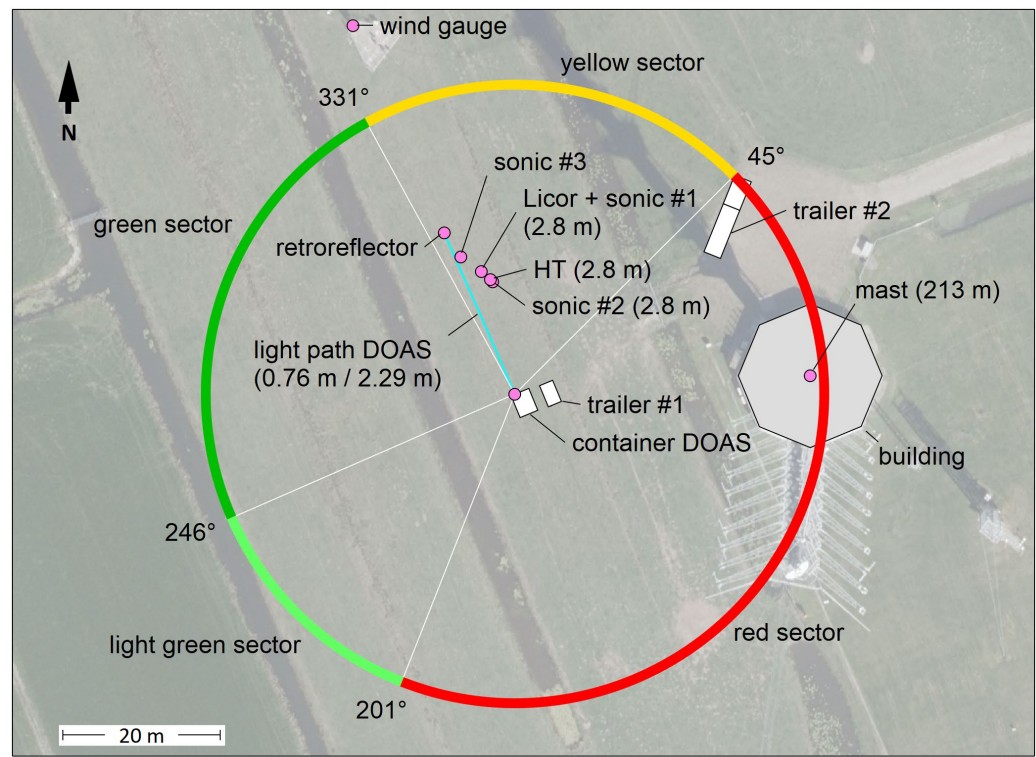

**Figure 1.** The Cabauw site with the location of the instruments. The coloured circle denotes wind origin sectors which are used for filtering data (see text). The green and light-green sectors indicate wind directions with minimally obstructed flow. Wind from the yellow sector is somewhat obstructed. Wind from the red sector experiences severe obstruction due to the building at the foot of the tall mast, the trailers and the DOAS container. Background aerial photo from opendata.beeldmateriaal.nl (downloaded 22-02-2022).

## 2.2 Instruments overview

For this campaign, the following instruments were set up in a field next to the 213 m mast at Cabauw. The two miniDOAS $NH_3$ instruments were placed above each other in a small container (see below for a detailed description of these instruments). The 22.1 m optical paths were directed at 336°, parallel to the ditches between the fields. The bottom path was at 0.76 m and the top path 2.29 m above the field. Anticipating prevailing winds from the south-west, the other instruments were positioned 3 m east of the miniDOAS optical paths (Figure 1), to minimise distortion of the incoming airflow. The HT8700E open-path $NH_3$ analyser (see below for a detailed description of this instrument; hereafter referred to as 'HT'), was mounted on a steel mast with the centre of its optical path at 2.80 m above the ground. On a second steel mast, 1.5 m from the first, a sonic anemometer (sonic#1; model Gill WindMasterPro™, Gill Instruments, Lymington, UK) was mounted. This sonic measured the 3D wind components at 32 Hz 2.8 m above the ground. The 10 Hz open-path $H_2O$ and $CO_2$ analyser (LI-7500DS, LI-COR Biosciences, Lincoln, USA) was placed at 2.83 m above the ground next to sonic#1.

From September 30 onwards, to evaluate the impact or sensor separation between the HT and the sonic#1 on the calculated $NH_3$ fluxes, a second sonic anemometer (sonic#2, model Gill WindMaster™, Gill Instruments, Lymington, UK) measuring at 32 Hz was installed 40 cm from the HT analyser. A third sonic, #3, was placed from the beginning of the campaign along the path but its data were not used in the final analysis. Because its presence may still influence the wind across the DOAS path it is included in Figure 1.


Data storage, power supplies, and remote control apparatus for these instruments were housed partly in the DOAS container and partly in trailer #1 parked next to this container. During part of the campaign, a large trailer#2 with additional instruments

was parked to the north-north-west of the 213 m mast.

In Figure 1 we show different coloured wind sectors. The selection is based on objects on the site that influenced the wind field and thus the flux intercomparison. The four wind sectors (Figure 1) were:

a)   The green sector (246°−331°): minimal disruption. Only the drainage ditches are expected to influence the wind field.

b)   The light green sector (201°−246°): minimal disruption. We expected the DOAS container to have some influence.

c)   The yellow sector (331°−45°): some disruption. The masts with HT and the sonics disturbed the wind field at the DOAS paths. At times, the sheep farmer positioned a small trailer there on the field to the north of the 213 m mast, and sheep were grazing there. This would have affected all instruments.

d)   The red sector (45°−201°): severe disruption. The 213 m mast, the building at the foot of this mast, the trailers and the

DOAS container all affected all instruments.


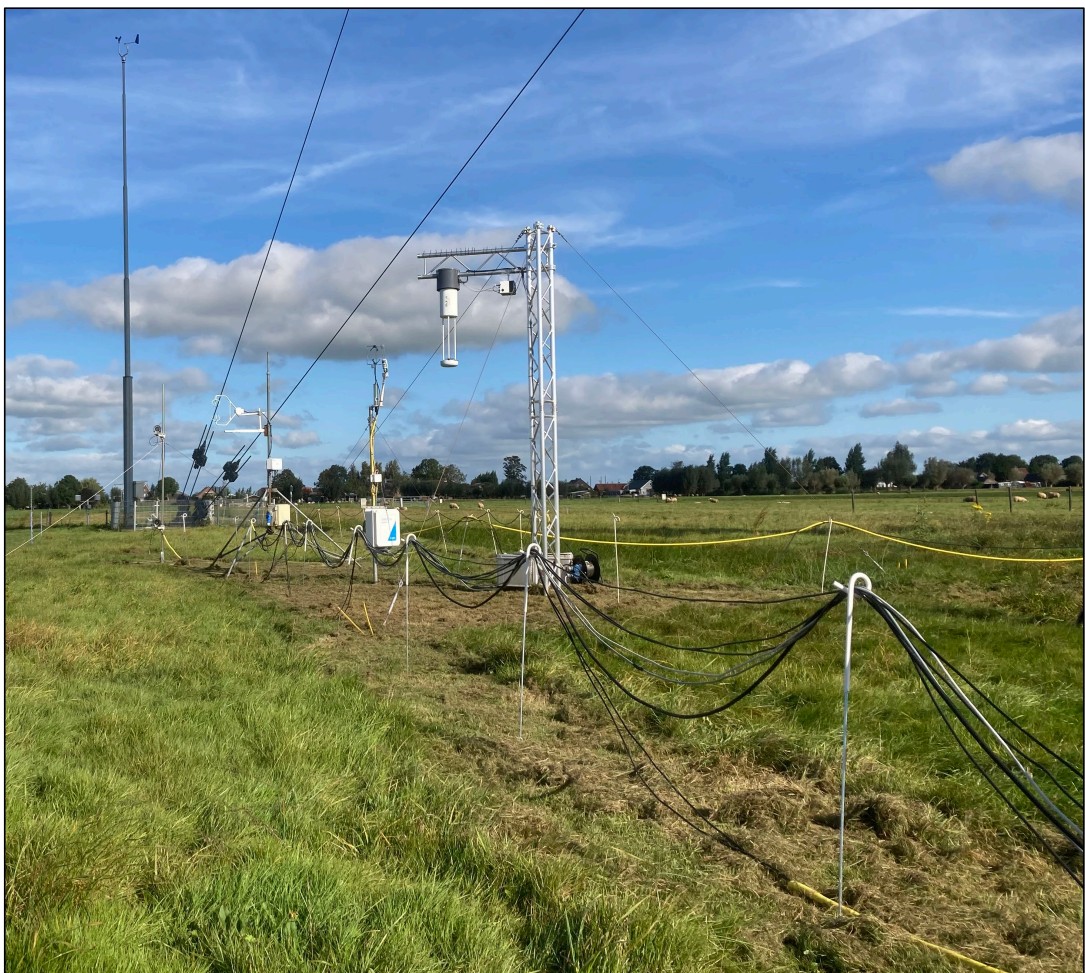

**Figure 2. The instruments, seen from the miniDOAS container looking north. From left to right: 10 m wind vane mast, mast with the two retroreflectors of the miniDOAS instruments, mast with sonic#3, mast with sonic#1 and LI-7500DS; mast with HT8700E and its cooling unit. Sonic#2 was placed later at 40 cm on the southeast side of HT8700E on the same mast (not shown in the photo). The 213 m mast is off to the right (east).**

### 2.3    Weather conditions

The Cabauw site has a mean annual air temperature of 10.4 °C and a mean annual precipitation of 770 mm. The thirty-year average temperature for September is 14.8°C, and the total precipitation is about 84 mm (KNMI, https://www.knmi.nl/klimaat-viewer, last access 12 May 2022). Historically, winds from the southwest tend to be most common in September. During the campaign, the weather was slightly warmer and substantially drier than normal for this time of year (Homan, 2021) (Figure S2). However, there were no extreme hot or cold spells. No significant precipitation occurred during the measurement period, except for a few short shower events late September. The wind direction during the campaign was variable and therefore different from the expected predominant wind direction.



## 3    Methods

### 3.1    Aerodynamic gradient method (AGM) NH₃ fluxes


#### 3.1.1    MiniDOAS instruments

DOAS, short for differential optical absorption spectroscopy, is an optical technique to measure trace gas concentrations over an open path in the atmosphere (e.g. Platt and Stutz, 2008). An open-path technique avoids the delay effects, reduced temporal resolution and interference from aerosols that result from $NH_3$ sticking to inlet lines, air filters and other surfaces in an

instrument (Parrish and Fehsenfeld, 2000). These qualities are especially relevant for $NH_3$ measurements in open air.

For this experiment, two identical RIVM miniDOAS 2.2D instruments were used. These are active DOAS systems, i.e., equipped with their own light source rather than using sunlight. The light is sent to a retroreflector over an open path of 22.1 meters and received back (Figure 3). Path-averaged $NH_3$ concentrations are retrieved from spectra taken in the 200–230 nm wavelength range.




**Figure 3. MiniDOAS setup in the field, using two instruments at different heights above the ground. The NH₃ flux is determined from the observed concentration difference between top and bottom paths and the turbulence measurements of sonic#1.**

The 2.2D instruments are a modified and further developed version of the miniDOAS 1.x described earlier (Berkhout et al., 2017; Volten et al., 2012b). MiniDOAS 1.x instruments have been operating in the Dutch national air quality monitoring network (LML, Landelijk Meetnet Luchtkwaliteit, https://www.luchtmeetnet.nl/, last access date: 20 Apr 2022) since 2016, providing hourly concentration measurements of $NH_3$ concentrations in ambient air at six locations in the Netherlands. The uptime of these instruments in 2021 was above 95% of the hourly values.


Improvements in the 2.2D version include the use of a more sensitive charge-coupled device detector and several optical components with higher reflectivity and/or transmission in the wavelength range used, leading to about a factor of 5 increase in optical throughput. The optical layout was simplified and an optical scanner was added, making the system less sensitive to small alignment changes. These modifications resulted in a substantial increase in precision and stability of the measurements,

as was needed for the monitoring of dry $NH_3$ fluxes with the AGM method. We aim to describe the miniDOAS 2.x in more detail in a forthcoming publication, in combination with the implementation of this version in the Dutch national air quality monitoring network LML.

For the AGM measurement, two miniDOAS 2.2D instruments were installed in an air-conditioned container and placed on a

stable metal frame attached to the container wall. Optical paths through quartz windows were located at 0.76 m and 2.29 m height, parallel to the local soil surface. The instruments each integrated spectra during 4 minutes and provided simultaneous path-averaged concentration values at 4-minute intervals. These concentrations were then averaged to 30-minute values.





### 3.1.2    MiniDOAS calibration and intercalibration

As the AGM method depends on the ability to measure small concentration differences between two heights, great care must
be taken to calibrate the two miniDOAS instruments properly, first individually and then as a pair, and to maintain this
calibration over the flux measurement period. This process is described below.

#### 3.1.2.1    Initial individual lab calibration

After assembly or maintenance, the individual miniDOAS instruments were calibrated according to the procedures used in the
LML-network. This included the acquisition of a 'reference spectrum' with known, preferably zero, concentrations of $NH_3$,
nitrogen oxide, and sulphur dioxide. For this, the zero-tunnel calibration facility at RIVM was used. This spectrum served as
a common reference for all measurements. We also acquired 'calibration spectra' of the three gases mentioned, using a flow
cell in the light path in combination with the zero-tunnel facility. These spectra contain the spectral fingerprint and cross-
section of these gases, used in the analysis. For this step, calibration gases of these components with a supplier-indicated
accuracy of 2% were used.

Afterwards, the accuracy of the $NH_3$ calibration was tested by providing two $NH_3$ mixtures in nitrogen from certified reference
cylinders, representing a low and a high concentration of about 30 mg m$^{-3}$ and 300 mg m$^{-3}$ in the atmosphere, respectively
(Certified Reference Materials, produced by the Dutch metrological institute VSL). These reference cylinders have a certified
accuracy of 3% and 2% respectively. The instrument calibration was considered valid if the measurement result was within
3% of the certified reference.

#### 3.1.2.2    Additional intercalibration for deposition

While suited for concentration monitoring, the calibration approach above is not precise enough for AGM, where concentration
differences of 0.1 µg m$^{-3}$ or better need to be determined, i.e., well below the 1% level. For this, an additional calibration of
the two miniDOAS instruments is needed, as a pair. This was done after installation in the field.
The instruments were manually set to a different alignment position, as indicated in Figure 4, the so-called cross-position. To
achieve this, a miniDOAS instrument pivots as one piece on a ball joint located near the quartz windows. As both instruments
now sample on average the same height region, results should be identical for all flow situations where the $NH_3$ gradient is
homogeneous over the horizontal path. In this cross-setting, the instruments were set to run for several days, until a sufficient
amount of variation in outside air concentrations were encountered. Typically, the intercalibration lasts at least 3 days under
ideal weather conditions (wind from an unobstructed direction, sufficient turbulent mixing).

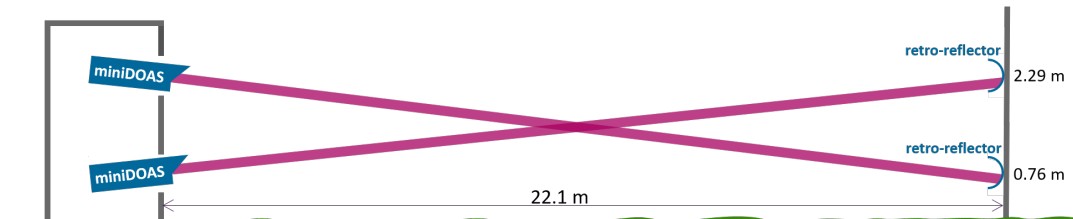


**Figure 4. MiniDOAS setup in the field, using two instruments in the cross-position. These zero-difference measurements are used
for the very precise intercalibration needed for flux measurements (see text).**

First, new simultaneous reference spectra for both instruments were obtained from the dataset obtained during the
intercalibration, to replace the reference spectra obtained in the zero-tunnel. The obtained absolute concentration values from
these spectra will be less accurate, i.e., they may have a small but fixed offset to the lab. They will however be more precise.



This is a useful step, as for the flux measurement the concentration difference between the two systems needs to be determined with minimum random error. Next, the spectra obtained in the cross period were processed with these new reference spectra. When comparing the results from both instruments in a scatterplot, the minor additional corrections to the offset and span can be obtained that are needed to make the instruments match perfectly, with offset 0 and slope 1.

In the Results section (Sect. 4.1), it will be illustrated that after these steps the pair was capable of measuring $NH_3$ differences well below our target precision of 0.1 µg m$^{-3}$. The new field reference spectra and the small additional corrections obtained in the cross-position are kept and also applied in the analysis of the flux measurements obtained in the parallel-position.

### 3.1.3 Flux calculation

The 30-minute concentration measurements obtained at the two measurement heights were combined with 30-minute averaged
transfer velocities to obtain the AGM $NH_3$ flux $F_{AGM}$, (e.g. Trebs et al., 2021):

$$F_{AGM} = -\frac{ku_*}{\ln\left(\frac{z_2}{z_1}\right) - \Psi_H\left(\frac{z_2}{L}\right) + \Psi_H\left(\frac{z_1}{L}\right)} \times [c_{NH_3}(z_2) - c_{NH_3}(z_1)] \qquad \text{Eq. 1}$$

Where $u_*$ is the friction velocity; $k$ is Von Kármán constant (0.4), $c_{NH_3}(z_n)$ is the $NH_3$ concentration at height $z_n$ and $z_1$ and $z_2$ are the heights of the bottom and top miniDOAS paths above the displacement height $d$ (assumed 2/3 of the canopy height), respectively. $\Psi_H\left(\frac{z}{L}\right)$ is the integrated stability function for heat, which is assumed to be the same for $NH_3$. $L$ is the Monin-Obukhov length. For unstable conditions ($L < 0$), we used the functions of Dyer (1974) and Paulson (1970). For stable
conditions ($L > 0$), we used the function of Beljaars and Holtslag (1991). Micrometeorological parameters $u_*$ and $L$ were calculated using EddyPro software version 7.0.6 (LI-COR Biosciences, Lincoln, USA) using data collected using sonic#1. AGM fluxes were calculated using custom software written in R. We follow the sign convention where positive fluxes indicate emissions and negative fluxes deposition.

### 3.2 Eddy covariance (EC) $NH_3$ fluxes

#### 3.2.1 HT8700E instrument

The open-path QCL-based $NH_3$ analyser (Healthy Photon Lt. Co., Ningbo, China, Model HT8700E; hereafter HT) was used to measure $NH_3$ concentrations at 10 Hz using the wavelength modulation spectroscopy technique. Technical details of the analyser have been described in Wang et al. (2021). The QCL sends a beam at 9.06 µm into an open-air Herriott cell which has two concave mirrors of high purity molybdenum with a coating that should withstand frequent cleaning with organic
detergents. The temperature of the QCL and detectors are stabilized by Peltier thermo-electrical coolers (TEC). The analyser is coupled to a compact external water and ethylene glycol chiller (Wang et al., 2021).

The HT8700 performance in laboratory and field experiments have been presented by Wang et al. (2021, 2022). The uncertainty of the $NH_3$ concentration measurements was estimated to be ±15% by comparing two commercially available high-sensitivity $NH_3$ analysers G2103 (Picarro Inc., Sunnyvale, USA) and EAA-911 (Los Gatos Research (LGR), San Jose, USA)
in the lab (Wang et al., 2021). Instrumental noise was first reported as 0.30 ± 0.046 ppbv after a one-week deployment in a subtropical rice paddy field in Southern China, with a flux detection limit of 7.1 ± 1.1 µg N m$^{-2}$ h$^{-1}$ (equivalent to 2.4 ± 0.4 ng $NH_3$ m$^{-2}$ s$^{-1}$). In the follow-up study (Wang et al., 2022), a slightly higher noise ratio (0.41 ± 0.06 ppbv) and flux detection limit (9.6 ± 1.5 µg N m$^{-2}$ h$^{-1}$, equivalent to 3.2 ± 0.5 ng $NH_3$ m$^{-2}$ s$^{-1}$) were found after one-month long monitoring at a wheat field in Northern China. The HT was calibrated in the factory using a glass tube that enclosed the open optical cell. Zero
calibration was done by flushing the cell with pure nitrogen. For span calibration, a $NH_3$ permeation tube and dilution system (KIN-TEK Analytical, Inc., La Marque, USA) was used. The permeation tube released a fixed rate of $NH_3$ at stabilized temperature. Different $NH_3$ concentrations in $N_2$ were provided with the diluter to the cell inserted into the instrument. The





off-factory calibration points of our instrument had rather larger values ($0 - 450$ µg m$^{-3}$) than typical field concentrations at the Cabauw site, but good linearity was obtained throughout the lab calibration range according to the manufacturer.

Raindrops, dust, and other contaminants on the mirrors (particularly the bottom one), cause light scattering which is shown in the optical signal strength (OSS) of the HT (Wang et al., 2021). In contrast to Wang et al. (2021, 2022) in this experiment we used an upgraded HT version being equipped with an automated mirror cleaning system (the SPIDER®) that can be activated remotely, which significantly reduced the manual cleaning burden. During this campaign whenever the OSS value dropped below 40% the lower mirror was cleaned using the SPIDER® that was activated remotely for 1 to 2 minutes at a time. In

addition, both mirrors were manually cleaned 1-2 times per week using lens tissue drenched in methanol if automatic cleaning was not sufficient. However, the OSS values gradually decreased over the experimental period especially after multiple rain events before the end of the campaign (Figure S3).

### 3.2.2    Flux calculation

The EC NH$_3$ fluxes and other micrometeorological parameters were calculated using EddyPro software (version 7.0.6, LI-
COR Biosciences, Lincoln, USA) at 30-minute intervals using the 10 Hz 'raw' data. The general flux calculation procedure followed the standard FluxNet methodology (Mcdermitt et al., 2011) and some basic settings are following Wang et al. (2021). For detailed settings and parameters of this study see Table S1. In addition to the analysis in EddyPro, additional spectral analyses were further tested to study the impact of high-frequency spectral damping and sensor separation on the flux results.

#### 3.2.2.1    High frequency spectral losses correction
The eddy flux method evaluates the vertical transport of gas, heat or momentum caused by a composition of turbulent eddies that cover the spectrum from cm to km scale or, in the time domain, from 10 Hz to 30 min scale. Measured EC fluxes correlate the vertical wind and the concentration variation, the covariance which can be visualised in a cospectrum showing the contribution of the large and small turbulent motions. The raw measurement data need corrections for turbulence-spectral

losses both in the low (> minutes) and high (> 1 Hz) -frequency range. For the open-path system, the former is caused by the finite averaging time, as the measurement system will not "see" large scale eddies that take longer than the 30-minute evaluation interval. The concentration changes that occur with a high frequency (linked to small eddies) are damped due to the sensing volume of the instrument (which is 50 cm high and will not show eddies that are 10 or 5 cm in diameter) and due to the spatial separation between sonic anemometer and gas analyser (Moore, 1986).

Using the EddyPro software, low-frequency flux losses were corrected according to Moncrieff et al. (2004). For estimating high-frequency flux losses for the open-path analyser, the theorical method from EddyPro (hereafter referred to as TEO; Moncrieff et al., 1997) was applied first. Two remarks have been made on this procedure. First, a difference can occur between the measured cospectra and the theoretical frequency distribution of Kaimal cospectra (Moncrieff et al., 1997; Kaimal et al., 1972). Second, in EddyPro's implementation of Moncrieff et al. (1997) the correction for sensor separation is independent of

the wind direction, which holds as long as the distance between the sonic anemometer and gas analyser is relatively small (Moore, 1986). Moore (1986) already indicated that in doing so the flux correction would probably be overestimated.

Therefore, to better understand the real field condition and equipment separation results in EC flux, an empirical approach using measured gas flux cospectra and sensible heat cospectra as reference was applied similar to Wintjen et al. (2020, hereafter referred to as ICO after 'in situ cospectral method')). In short, a cospectrum shows how much each frequency (eddy) contributes

to the flux. A fitting of the normalised temperature based on the trace gas cospectra using an empirical transfer function shows how well the gas analysers can resolve the contribution of eddies in the high-frequency range of the gas flux cospectra and therewith suffer from damping effects that are assumed to be absent for temperature. We used the same empirical transfer function as Wintjen et al. (2020) and made the non-linear fit for frequencies higher than 0.1 Hz. For further details on the cospectral correction method, we refer to Sect. 2.3.2 of Wintjen et al. (2020). To ensure the quality of the half-hourly damping





correction factors, values were filtered according to the following criteria: QC flag 0 in Eddy Pro (Mauder and Foken, 2006); $u_* > 0.1$ m s$^{-1}$; the variances in the NH$_3$ concentration are below 2 times the standard deviation plus the mean of the overall campaign period; EddyPro found a maximum in the covariance within the prescribed time lag window. In total, 661 half-hour NH$_3$ correction factors were left for correcting their respective fluxes. If half-hourly corrections factors were not available due to quality criteria, we used daily medians to correct corresponding fluxes.


### 3.2.2.2  Modified WPL correction

Open-path trace gas concentrations are affected by density variations in the up- and down going air movements. The Webb-Pearman & Leuning (WPL) correction accounts for that. Two WPL methods were used. First, the classic WPL method was used to correct H$_2$O measurements from LI-7500DS (Webb et al., 1980) part from that, the NH$_3$ flux is also affected by

spectroscopic effects (Burba et al., 2019). The spectroscopic part is instrument dependent and deals with the effect of changing H$_2$O concentrations and their impact on the absorption line used for NH$_3$. Hence, the modified WPL method was applied to correct the HT-measured NH$_3$ flux following Wang et al. (2021):

$$F_{EC} = A\left[\overline{w'\rho'_A} + B\mu\frac{\overline{\rho_A}}{\overline{\rho_d}}\overline{w'\rho'_v} + C\left(1 + \mu\frac{\overline{\rho_v}}{\overline{\rho_d}}\right)\frac{\overline{\rho_A}}{\overline{T_a}}\overline{w'T'_a}\right] \qquad \text{Eq. 2}$$

where $\rho_A$ is the NH$_3$ density corrected for temperature (see Sect. 4.1.2), $\rho_d$ is the dry air density, $\rho_v$ is the water vapor density, $\mu$ is the molar mass ratio of dry air to water vapour, $\overline{w'\rho'_v}$ is the water vapour flux measured by the LI-7500DS, $T_a$ is the air temperature and $\overline{w'T'_a}$ is the sensible heat flux from the sonic anemometer. $A$, $B$, and $C$ are dimensionless parameters accounting for the spectroscopic effects, which vary with ambient temperature, pressure and water vapour content (Wang et al., 2021).

### 3.3  Quality control and filtering

The quality control of the AGM and EC NH$_3$ fluxes was proceeded as follows. Observations from the HT were filtered out before the EC flux analysis if the optical signal strength (OSS) of the NH$_3$ analyser was below 40% (Figure S3). Firstly, after EC analysis in EddyPro was completed, EC fluxes were removed if a quality flag of 2 was assigned according to the stationarity and integral turbulence tests proposed by Mauder and Foken (2006). Secondly, both the EC and AGM fluxes with $u_*$ values

smaller than 0.1 m s$^{-1}$ were discarded, to filter out observations during low-turbulent mixing conditions. Thirdly, a moving window outlier filter was applied to the remaining EC and AGM fluxes, removing points if two times the standard deviation of the adjacent six flux values was exceeded (Wang et al., 2021; Wang et al., 2022). Finally, the data was grouped into 4 different wind sectors (green, light green, yellow and red) as described in Figure 1. Only observations from the green and light green sectors were used for the intercomparison of the AGM and EC fluxes. An overview of the applied filters and the

percentage of accepted fluxes per filter step are shown in Table S2 in the supplementary materials.

### 3.4  Uncertainty analysis

The random error of the half-hourly AGM NH$_3$ fluxes ($\sigma_{F_{AGM,}}$) from the miniDOAS instruments has three error components and was estimated as follows:

$$\sigma_{F_{AGM}} = \left|F_{NH_3}\right|\sqrt{\left(\frac{\sigma_{u_*}}{u_*}\right)^2 + \left(\frac{\sigma_{c_{NH_3}(z_2) - c_{NH_3}(z_1)}}{c_{NH_3}(z_2) - c_{NH_3}(z_1)}\right)^2 + \left(\frac{\sigma_{f(z,\Psi)}}{f(z,\Psi)}\right)^2} \qquad \text{Eq. 3}$$

Here, the three error components are:



1. the relative error of the $u_*$ values, $\frac{\sigma_{u_*}}{u_*}$,

2. the relative error of the difference in the miniDOAS NH$_3$ concentration at height $z_1$ and $z_2$, $\frac{\sigma_{c_{NH_3}(z_2) - c_{NH_3}(z_1)}}{c_{NH_3}(z_2) - c_{NH_3}(z_1)}$,

3. and an error term related to the stability correction at each measurement height, $\frac{\sigma_{f(z,\Psi)}}{f(z,\Psi)}$, with

$$f(z, \Psi) = \ln\left(\frac{z_2}{z_1}\right) - \Psi_H\left(\frac{z_2}{L}\right) + \Psi_H\left(\frac{z_1}{L}\right).$$

The relative errors of the $u_*$ are estimated in EddyPro following Finkelstein and Sims (2001). The relative error of the miniDOAS NH$_3$ concentration differences is determined to be 0.088 μg m$^{-3}$ during the cross periods (see Sect. 4.1.1). Finally, we assumed that the errors in the height of the measurements ($z_1$ and $z_2$) were negligible and that the error in $f(z, \Psi)$ solely depend on the errors in the stability corrections. Following Wolff et al. (2010a), we assumed that the stability corrections have a relative error of 10%.


The random error of the half-hourly uncorrected EC NH$_3$ fluxes is estimated in EddyPro following Finkelstein and Sims (2001). To determine the random error of the WPL corrected NH$_3$ fluxes, Eq. 2 is rewritten as a sum of four terms $F_1$, $F_2$, $F_3$ and $F_4$:

$$F_{EC} = A\overline{w'\rho_A'} + AB\mu\frac{\overline{\rho_A}}{\overline{\rho_d}}\overline{w'\rho_v'} + AC\frac{\overline{\rho_A}}{\overline{T_a}}\overline{w'T_a'} + AC\mu\frac{\overline{\rho_v}}{\overline{\rho_d}}\frac{\overline{\rho_A}}{\overline{T_a}}\overline{w'T_a'} = F_1 + F_2 + F_3 + F_4$$

Eq. 4


The random error of the WPL method corrected NH$_3$ fluxes ($\sigma_{F_{NH_3,HT}}$) is computed as follows:

$$\sigma_{F_{EC}} = \sqrt{\sigma_{F_1}^2 + \sigma_{F_2}^2 + \sigma_{F_3}^2 + \sigma_{F_4}^2}$$

Eq. 5

Here, the term $F_1$ represents the NH$_3$ flux term in the WPL correction, whose random error ($\sigma_{F_1}$) largely follows the random error of the HT NH$_3$ fluxes ($\overline{w'\rho_A'}$) from EddyPro. The term $F_2$ represents the water vapour term, and $F_3$ and $F_4$ together are

the heat terms in the WPL correction, respectively. To determine the random error of each term, the relative errors of the included variables are propagated. The relative error of the NH$_3$ density ($\rho_A$) from the HT reported by the manufacturer is 15% (Wang et al., 2021). The relative errors of the dry air density $\overline{\rho_d}$, water vapor density $\overline{\rho_v}$ and air temperature $\overline{T_a}$ from sonic#1 are estimated to be 5%, 5% and 1%, respectively (Li-Cor Inc, 2022). Finally, the relative random errors of the water vapour flux ($\overline{w'\rho_v'}$) and the sensible heat flux ($\overline{w'T_a'}$) were taken from EddyPro.

**3.5     Footprint analysis**

The footprint of the EC fluxes showing the contributing area of measured fluxes was analysed following the method from Kljun et al. (2015). Inputs for this method include the EC measurement height ($z = 2.80$ m), roughness length (assumed to be 0.15 times canopy height), friction velocity ($u_*$), the Obukhov length, the standard deviation of the lateral wind ($v$) component, wind direction, mean wind speed, and the boundary layer height. Apart from the boundary layer height, other parameters were

measured by the EC system. The hourly boundary layer height data was obtained from Climate Data Store (CDS) source (ERA5 hourly data: https://cds.climate.copernicus.eu/cdsapp#!/dataset/reanalysis-era5-single-levels, last access date: 1 Feb 2022) and the hourly values were linearly interpolated to half-hourly values for the footprint calculation. The flux footprint prediction (FFP) method (http://footprint.kljun.net, last access date: 1 Feb 2022) was used for coding and plotting. Here,





footprints were only determined for EC NH₃ flux after quality filtering (see Sect. 3.3). No separate footprint analysis was done
for the AGM fluxes.

## 4    Campaign results

### 4.1    NH₃ concentrations

#### 4.1.1    MiniDOAS intercalibration

The lab calibration procedure of both individual miniDOAS instruments is described in the instrument section (Sect. 3.1.2),
with a validated accuracy of better than 3%. Here, only the results of the intercalibration of the two instruments in the field are
shown, which aims to increase the precision of the concentration difference measurement further.

With the instruments placed in cross-position, intercalibrations were taken in three periods: at the beginning and end of the
campaign and once during the campaign. In total, 35% of the 7-week uptime was spent for the intercalibration during the
campaign.

For intercalibration purposes, a concentration gradient is acceptable in these measurements, but this gradient needs to be the
same over the full length of the paths. Therefore, the data was filtered for well-mixed situations ($u_* > 0.1 \, \mathrm{ms^{-1}}$) and obstacle-
free wind directions (green and light green). Figure 5 shows a scatterplot of the obtained concentration measurements by both
instruments matching these requirements.

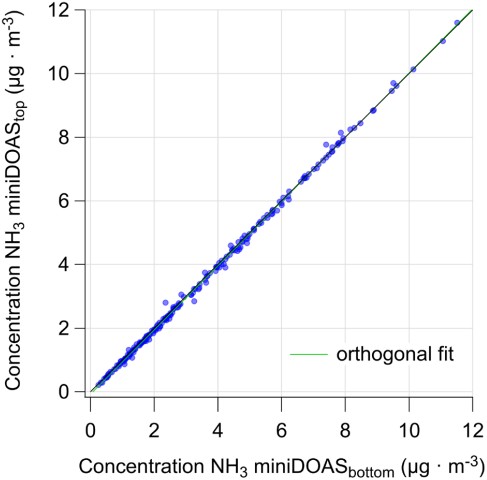


**Figure 5. Scatterplot of data obtained by the two miniDOAS instruments during all three cross-periods. Data was filtered to include
only obstacle-free wind directions and turbulent conditions ($u_* > 0.1 \, m \, s^{-1}$). Using an orthogonal fit, an offset of 0.07 ± 0.01 µg m⁻³ and a slope of 1.009 ± 0.002 (the green line) was found.**

With the obtained offset and slope between the systems, the concentrations of miniDOAS_top were corrected, in order to get a
scatterplot with offset 0 and slope 1 after correction. Next, the same minor correction factors were applied to the concentrations
of miniDOAS_top over the full campaign. The standard deviation of the residuals was used as an estimate of the remaining
random uncertainty in the concentration difference $c_{\mathrm{NH_3}}(z_2) - c_{\mathrm{NH_3}}(z_1)$ after correction. The random error of the miniDOAS
NH₃ concentration differences $\sigma_{c_{\mathrm{NH_3}}(z_2) - c_{\mathrm{NH_3}}(z_1)}$ in Eq. 3 was determined to be 0.088 µg m⁻³.




The concentration differences between the miniDOAS paths during the three cross periods are shown in Figure 6. The blue elements of the time series meet the selection criteria mentioned above. For these blue data points, statistics are given separately for the three periods.

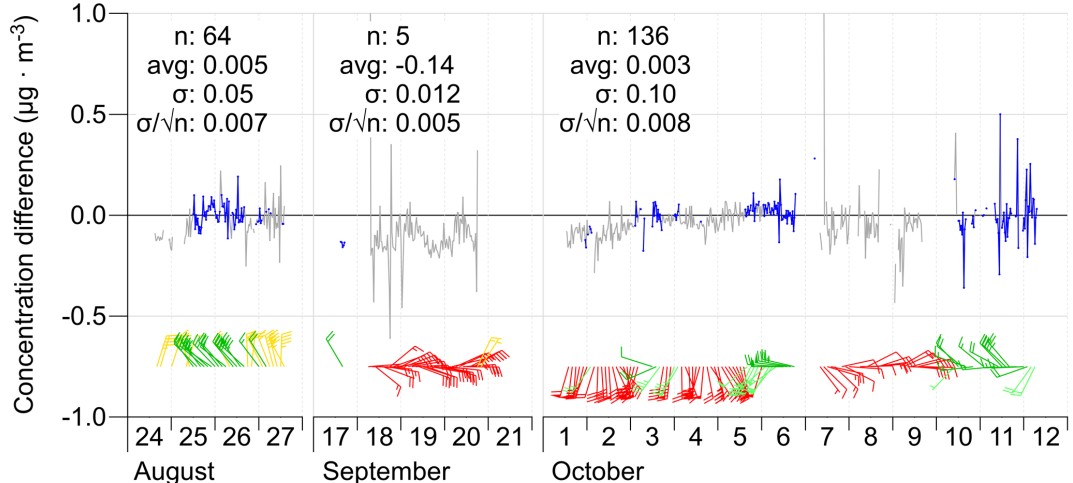

**Figure 6. Top trace: Time series of de observed NH₃ concentration difference between the two miniDOAS instruments during the three cross periods, after correction of the top miniDOAS values based on the intercalibration as described in the text. Only data during well-mixed conditions ($u_* > 0.1\ \mathrm{ms^{-1}}$) are shown. Measurements from obstacle-free wind directions are blue, other directions are grey. The sets of statistics given in the plots apply to the blue measurements only. Bottom trace: 30-minute wind vectors colour-coded with the wind sectors described earlier. Wind speed is indicated as barbs, as used on meteorological maps. To reduce clutter, only 1 in 4 wind vectors are shown.**

In the first cross-period, the average half-hourly difference was $0.005 \pm 0.007$ µg m⁻³ and the standard deviation was 0.05 µg m⁻³ ($n = 64$). For the last cross-period, these values were $0.003 \pm 0.008$ µg m⁻³ and 0.10 µg m⁻³ ($n = 136$), respectively. The average only shifted by 0.002 µg m⁻³ between cross-periods 1 and 3. This is well within the combined uncertainty range. The spread of the half-hourly values has increased from 0.05 in cross-period 1 to 0.10 µg m⁻³ in cross-period 3. This increase was likely at least partially caused by the gradual decay of the lamp intensity, causing a larger measurement error. The 'ageing' of the field reference spectrum may also have played a role here, but this was not studied further. The conclusion is that, over the full campaign period, the zero-level of the difference measurement has been stable, and the individual difference measurements showed a typical spread of 0.1 µg m⁻³ or less.

The second cross-period contained almost no valid data points, as the wind during this cross-period was coming from the direction of the largest obstacle of all: the 213-meter mast. Its impact on the analysis above was therefore almost zero. The data during this cross-period show how large the magnitude of the effect of upwind obstacles can be. The grey points taken during this cross-period differed systematically from zero by about 0.2 µg m⁻³, reflecting that the gradients were different at different locations along the paths.

### 4.1.2 HT concentration corrections

The HT NH₃ concentration measurement contained a considerable amount of gaps in the data (21% during the 5-week uptime). These gaps largely occurred during rain and mirror cleaning afterwards. At the start of the campaign, the HT instrument had an offset of about -7 µg m⁻³ (data not shown). After the campaign, the analyser was recalibrated in the lab and the 'zero' was found to be -6.3±0.3 µg m⁻³ when flushing pure nitrogen gas for 6 hours through the calibration cell while the temperature was kept constant at 17 °C. Before temperature correction, raw HT and miniDOAS_top's average difference was -5.3 µg m⁻³ (range -15 to 6 µg m⁻³, $n = 1180$) during the overlapping period of the campaign. The raw half-hourly HT NH₃ concentrations showed





inconsistent differences compared to the miniDOAS concentration levels, which varied with air temperature. After applying a third-order polynomial fit of the HT-miniDOAS concentration difference versus temperature, the corrected concentrations for HT were finally obtained (Figure S4). Tempature mainly impacted on the offset of its concentration and it seemed to have a negligible influence on the span of the HT's concentration (slope ≈ 0.97, Figure 7).

**4.1.3    Comparison miniDOAS and HT concentrations**

After application of the temperature correction on the $NH_3$ concentrations of the HT, the concentration of the two instruments were very similar ($R^2 = 0.97$, Figure 7). Furthermore, the time series of the corrected $NH_3$ concentrations from both instruments captured the same temporal pattern and peak events. The highest concentrations were observed by both systems at noon when air temperature reached the highest level of the day (Figure 8).

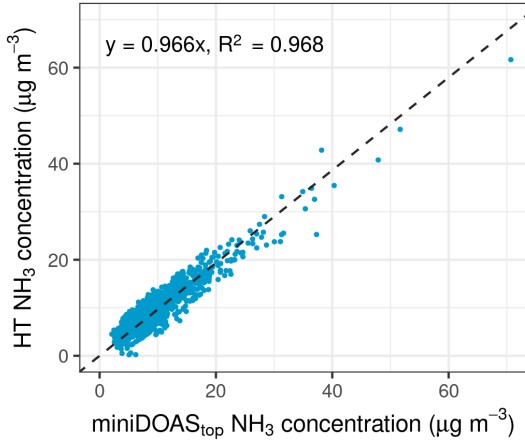


**Figure 7 Scatterplot of the $NH_3$ concentrations from the miniDOAS_top and the temperature corrected HT instrument during parallel measurements (correlation line was forced through the origin).**

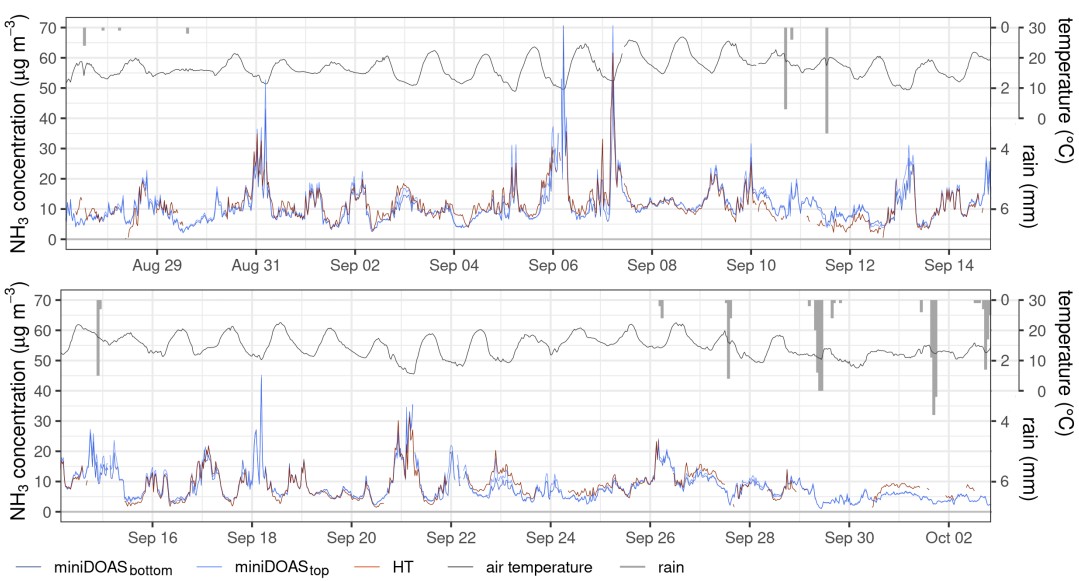

**Figure 8. Time series of the measured, unfiltered $NH_3$ concentrations after temperature correction from the HT (red) and the miniDOAS instruments (dark blue for the bottom one; light blue for the top one) in µg m⁻³, the hourly ambient temperature (black) in ⁰C and the amount of rainfall (grey bars) in mm.**



### 4.2 Uptime, filtering and quality control

For the AGM method, the vertical $NH_3$ concentration gradient measured by the miniDOAS instruments and the transfer
velocity from the sonic#1 anemometer were used to determine the $NH_3$ flux. Figure S5a in the supplementary materials shows
the full time series of the $NH_3$ flux derived using the miniDOAS setup. The miniDOAS setup had an uptime of nearly 100%
over the full campaign (1142 hours). Except for the 35% intercalibration period, 80% of the remaining parallel measurements
(597 hours) were left after filtering out low turbulent mixing conditions ($u_* < 0.1\ \mathrm{m\ s^{-1}}$) and outliers. For the EC $NH_3$ flux
measurements, Figure S5b shows the full time series. The uptime of the HT instrument was 79% during the 5-week field
operational period (685 hours). After filtering for fluxes with poor quality flags, $u_* < 0.1\ \mathrm{m\ s^{-1}}$ and outliers, 59% of the valid
observations remain (516 hours). At last, after filtering all measurements on September 11th (discussed in Sect 4.3), 424
overlapping hours were left for flux comparison between two instruments.

### 4.3 Comparison of the AGM and EC fluxes

The $NH_3$ fluxes from the two methods are shown in Figure 9. Here, the EC fluxes corrected for flux damping in EddyPro are
shown, which is considered as a reference method. Observations on the 11th of September were excluded due to remarkable
large differences between the measured fluxes on that day, although they originated from green wind directions. We assume
this was related to manuring at the adjacent field that might have disturbed the footprint homogeneity of the flux but we have
no evidence to support that. After quality control filtering, the EC and AGM fluxes have a similar range and pattern. Within
the green and light green sectors, the highest $NH_3$ emission measured with the AGM setup was 0.18 µg m$^{-2}$ s$^{-1}$ and deposition
was 0.15 µg m$^{-2}$ s$^{-1}$. The highest observed $NH_3$ emissions with the EC setup was 0.16 µg m$^{-2}$ s$^{-1}$ and deposition was 0.10 µg m$^{-2}$ s$^{-1}$.

At the start of the measurement period, the AGM and EC fluxes were quite different. During the first days, the miniDOAS
system presented $NH_3$ deposition, while the HT showed $NH_3$ emissions. In this period, the prevailing winds were from the
north/northeast, categorised as yellow (see Figure 1), where sheep were occasionally located upwind of the instruments. This
may have caused inhomogeneity of the source/sink pattern within the footprint area (see below), which would have violated
the AGM/EC calculation assumptions. Furthermore, the $NH_3$ concentrations during this episode were relatively high as
manuring activities were still allowed until 15 September on the grasslands surrounding the measurement site. The $NH_3$ fluxes
from the two methods compared well during the second half of the measurement campaign during times when the airflow was
unobstructed (green and light green wind categories).

Figure S6 in the supplementary materials shows cumulative fluxes using only data after September 15, when manure spreading
was not allowed anymore. Considering only high-quality measured fluxes during this period, the cumulative fluxes of the
AGM and EC were likely within the order of ~10%. The daily cumulative $NH_3$ fluxes illustrated that the differences between
the two methods can largely be traced back to September 24th. On this day, the daily cumulative AGM fluxes may have been
substantially larger than the cumulative EC flux.



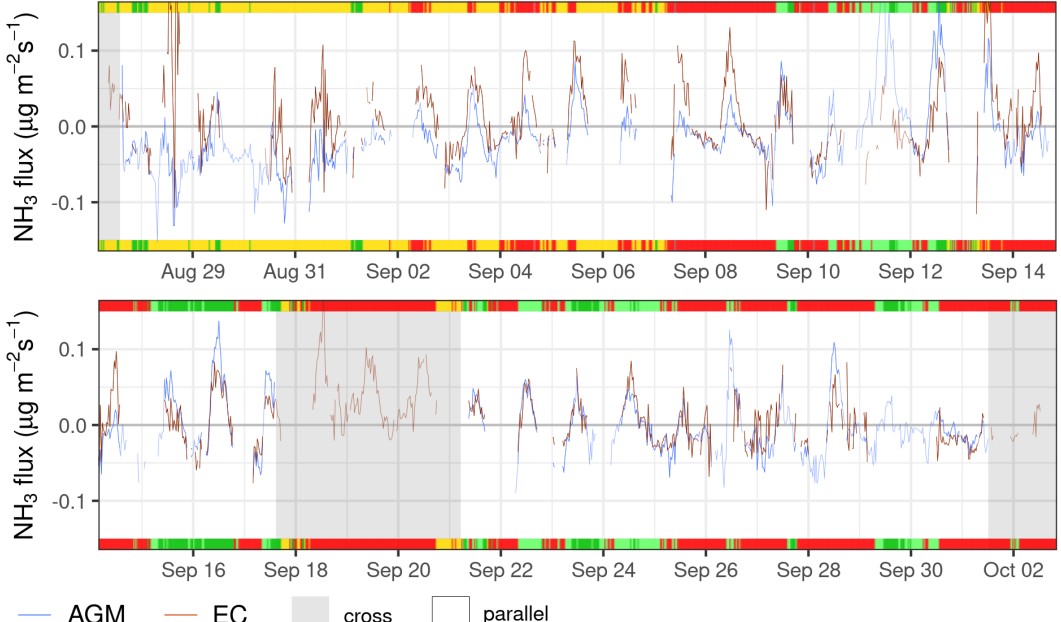

**Figure 9. Timeseries of the NH₃ fluxes of AGM with miniDOAS instruments (blue) and the EC method from the HT (red). Positive**
**fluxes indicate emissions, negative fluxes deposition. The colours in the upper and lower borders indicate the prevailing wind**
**directions from Figure 1. The intercalibration periods for the miniDOAS instruments are shown against a grey background. The**
**thick lines indicate the NH₃ fluxes that were left for intercomparison after all filters were applied.**

Figure 10 shows the comparison of the EC (EddyPro calculated) and AGM NH₃ fluxes per categorized wind direction. There

is a strong correlation ($r = 0.87$) between the EC and AGM NH₃ fluxes at times where the airflow was unobstructed, i.e.,

when the wind came from the directions categorized as green. In this category, the differences between the EC and AGM NH₃

fluxes were relatively small (RMSE = 0.027, bias = 0.012), too. There is a moderate correlation between the EC and AGM

NH₃ fluxes in the light green ($r = 0.71$) and the red categories ($r = 0.69$). In both the green and light-green categories, the

AGM based fluxes were approximately 30% above the EC based levels (slope = 1.3 (green) and slope = 1.35 (light green)). In

the red category, the airflow was partially obstructed by large objects (e.g., the 213 m mast, trailers and containers with

measurement devices). In this category, the EC fluxes were generally larger than the AGM fluxes (slope = 0.64), but relatively

small differences (RMSE = 0.034, bias = -0.016) between the EC and AGM NH₃ fluxes were found still. The poorest agreement

($r = 0.33$, RMSE = 0.072, bias = -0.045) between the two methods is found for the yellow wind direction category. In this

category, the HT often observed NH₃ emissions while the miniDOAS setup observed deposition of NH₃.



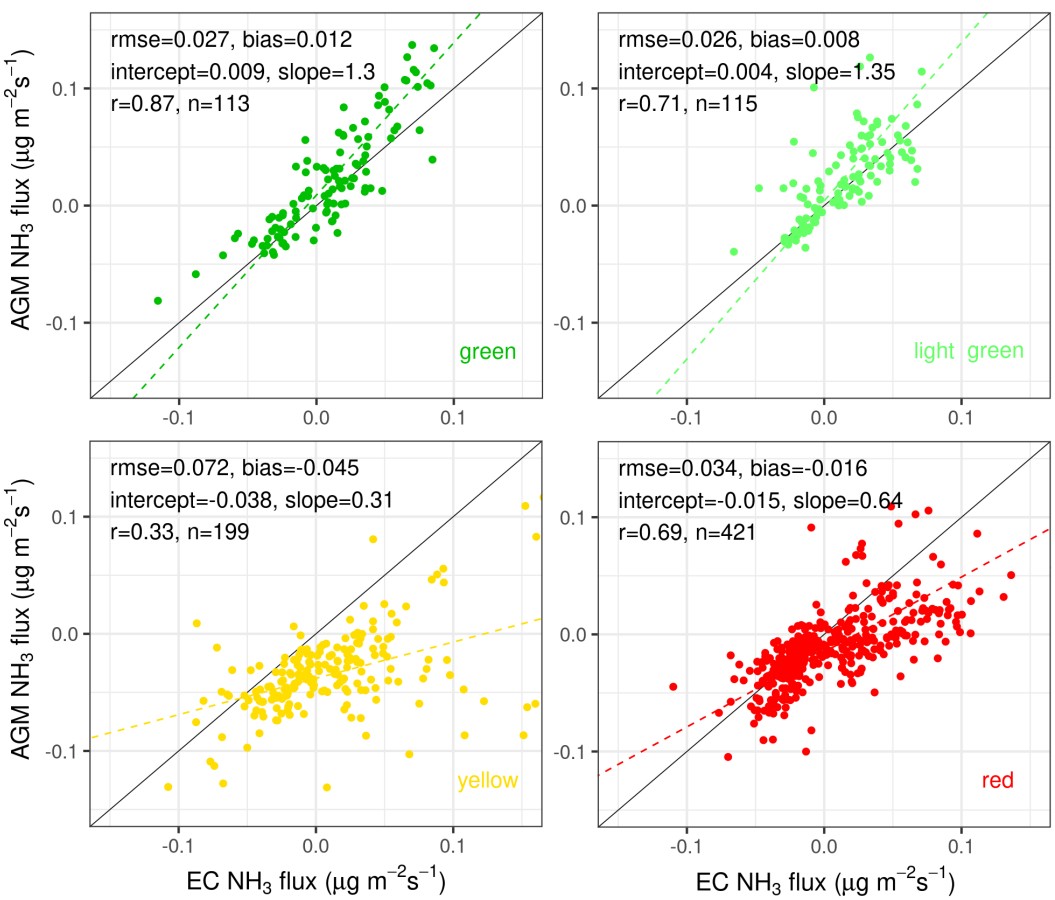


**Figure 10 Comparison of the AGM NH3 fluxes from the miniDOAS instruments and the EC NH3 fluxes from the HT per categorized wind direction (see Figure 1).**

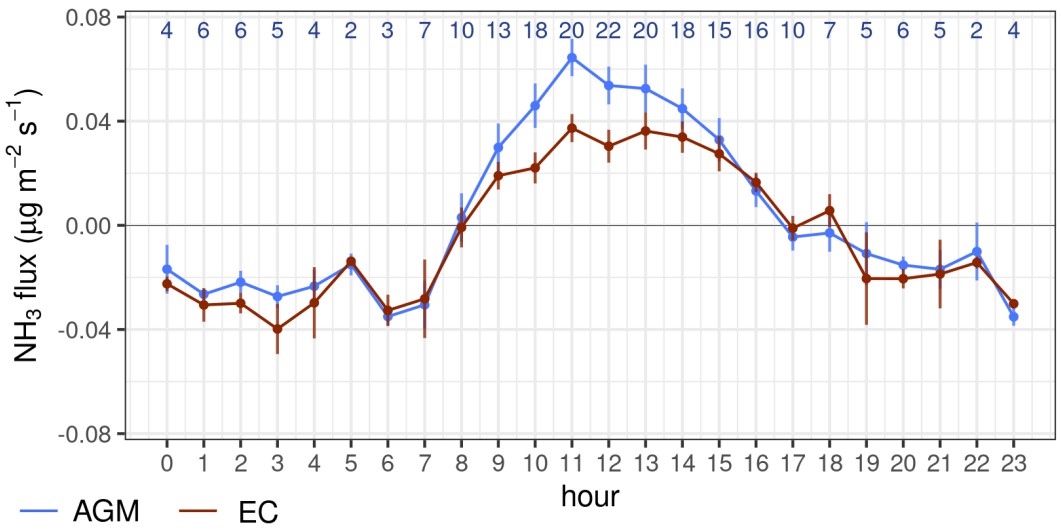





**Figure 11. Mean diurnal cycle of the EC and AGM NH$_3$ fluxes. Positive flux is emission, negative flux is deposition. The error bars indicate the standard error of the hourly means ($\sigma/\sqrt{n}$). The number of hours averaged are listed in blue text at the top. Here, filtered NH$_3$ fluxes from only the green and light green wind directions where both systems have a valid flux observation were used. Data from the 11$^{th}$ of September is excluded, too, due to a potential emission event causing footprint heterogeneity.**

The two methods showed a similar diurnal pattern using filtered NH$_3$ fluxes only from the green and light green wind directions (Figure 11). NH$_3$ was generally emitted during the day and deposited during the night. Between 10 am and 2 pm, the AGM

fluxes were a factor ~1.7 higher than the EC fluxes. Figure S7 in the supplementary materials shows the diurnal pattern using only data after September 15. The mid-day differences between the two are smaller, but still exist, even though manure spreading was not allowed anymore.

### 4.4 Uncertainty analysis

Figure 12 shows the random errors of the AGM and EC NH$_3$ fluxes and the contribution of different components to the error.

The random errors of the two showed a similar range of values. On average, EC NH$_3$ fluxes had a slightly lower estimated error. The mean random error of the AGM NH$_3$ flux was 9.8 ng m$^{-2}$ s$^{-1}$ (median 7.6 ng m$^{-2}$ s$^{-1}$), while the mean random error of the EC NH$_3$ fluxes amounted to 5.5 ng m$^{-2}$ s$^{-1}$ (median 4.1 ng m$^{-2}$ s$^{-1}$). The mean and median relative random errors were 87% and 23% for the AGM flux (excluding cross-periods) versus 61% and 15% for the EC flux.

The random errors of the AGM fluxes showed a clear diurnal pattern. During the daytime, the random errors were relatively large and peaked around noon, because the observed gradient was the smallest at this time. As a result, the measurement error in the NH$_3$ concentration differences from the miniDOAS instruments dominated. During the night, the random errors were relatively small. Here, the errors in the measured $u_*$ values from the sonic anemometer had a relatively large contribution to the total random error. The contribution of $u_*$ was relatively large when NH$_3$ was deposited at this site, i.e. the measured NH$_3$

deposition estimates were especially sensitive to the random error in $u_*$. The largest random errors in the NH$_3$ fluxes largely coincided with moments where the error in the stability correction takes over, i.e. when a substantial stability correction was applied to the measurement heights of the miniDOAS instruments. This occurred occasionally during night-time, usually around midnight. Compared to the random error of the AGM NH$_3$ fluxes of the miniDOAS instruments, the diurnal cycle of the random error in the EC NH$_3$ fluxes of the HT was less apparent. The contributions of the heat terms in the WPL correction

to the total random error were negligible. The contribution of the error in the WPL water vapour term can be quite substantial (max. ~75%) in incidental cases but is generally between zero and ~20% during daytime.

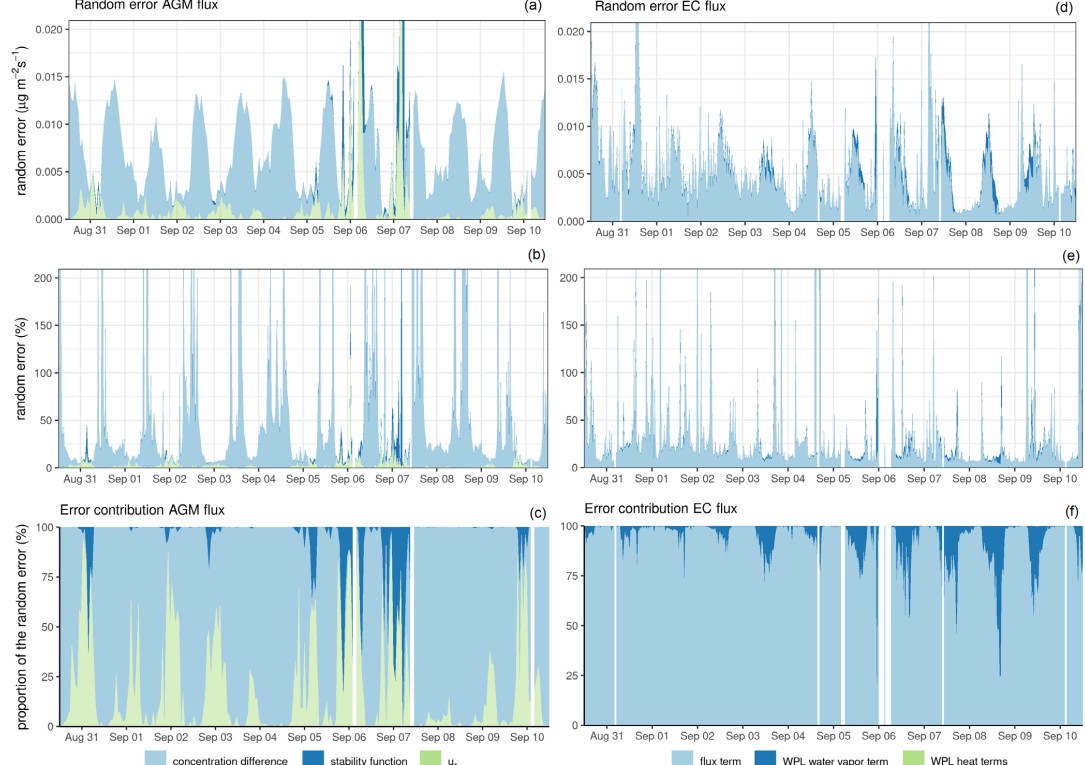

**Figure 12. The absolute and relative random errors and the corresponding error contributions (in % of the random error) of the AGM (a, b, c) and EC (d, e, f) NH₃ fluxes from August 31st to September 10th. For the EC fluxes, the light blue component (flux term) refers to F1 in Eq. 4, based on fluxes determined using EddyPro, taking into account the damping correction and term A from Eq. 4.**

### 4.5    Footprint analysis

The footprint of the EC NH₃ fluxes was computed at sonic#1's height using the method from Kljun et al., (2015) and shown in Figure 13. Overall, 80% of the flux originated from an area within approximately 100 meters distance from the measurement devices. Furthermore, the influence of the 213 m mast seems visible and reduces the footprint to the southeast. Because the highest measurement point has the largest footprint, the footprint of the miniDOAS instruments, especially miniDOAS_bottom, will be substantially smaller. The measured fluxes are assumed to be representative of the footprint area. The largest footprint area determines the outside perimeter of the area within which the landscape should be homogenous. If that is not the case it can be expected that the AGM and EC methods will end up with different results.





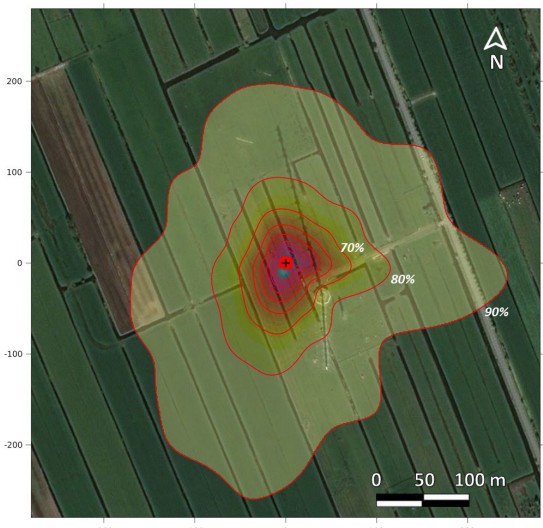

**Figure 13. Footprint climatology estimate of the EC measurements ($z = 2.80$ m). The red curves are at 10% footprint contour lines.**
**Background map data: © Microsoft, CNES Distribution Airbus DS**

### 4.6 Damping correction methods: TEO versus ICO for EC flux

To evaluate the effect of damping on the EC flux, both the theoretical method from EddyPro (TEO) and the empirical method (ICO) were used. In the results above we use the EddyPro theorical approach (Table S1) since we consider that is the 'standard' evaluation method. The comparison showed that theoretical flux correction factors in EddyPro are larger than the levels

obtained for the empirical (ICO) factors for $CO_2$, $H_2O$, and $NH_3$ (Figure S8). The first cause for that is that the sensible heat flux cospectrum indicates that the theoretical Kaimal cospectrum does not represent the turbulent characteristics of the sites well enough (Figure S9). If applying the ICO method on the HT data, the correlation with the AGM results gets worse in both the light green and green wind sectors (Figure S10). The ICO method seems to be conceptually better. However, ~50% of the dataset had to be corrected using daily median values, because the measurement-based ogives were noisy (caused by low flux

conditions). We therefore decided for this relatively short campaign to still use the TEO method for the flux comparison with AGM method.

The effect of the sensor separation on the damping was evaluated. We tried to split the effect of the sensing volume size and the effect of the 1.5 m sensor separation between the HT and the sonic#1 on the total flux damping during the entire campaign

period. This was done with the ICO method by comparing the damping factors when two sonics were both installed shortly at the site 2 days before the end of campaign. For sonic#1 data at 1.5 m from the HT instrument median flux damping of $NH_3$, $CO_2$, and $H_2O$ were 37%, 1% and 1%, respectively. For the second sonic anemometer, sonic#2 at 0.40 m of the HT instrument and 1.35 m to the LI-7500DS the empirical method gave 16%, 32%, and 31% for $NH_3$, $CO_2$, $H_2O$ damping. The effect of separation between the HT and the sonic#1 caused ca. 20% extra damping for $NH_3$ flux, while separating the LI-7500DS and

sonic#2 by the same distance caused 30% extra flux losses for both the water vapor and $CO_2$ flux. That could be explained because the HT has a 4 times longer vertical path length than the LI-7500DS sensor (0.50 m *vs.* 0.125 m) in which higher frequencies will be damped anyway.

To further check whether the TEO method can properly correct the sensors separation impact during the entire campaign period, especially between HT and sonic#1, this method was applied to compare the damping effect on both sonics in the short

period as well. With the TEO method, median damping factors of $NH_3$, $CO_2$, and $H_2O$ were 41%, 14% and 14% using sonic#1 as reference and 20%, 38%, and 37% using sonic#2. The TEO method produced the same damping difference between sonic#1





and sonic#2 for $NH_3$ flux as the ICO method (*ca.* 20%). After implementation of the damping correction, the fluxes obtained simultaneously with the HT and sonic#1 or sonic#2 were the same (see Figure S11). However, by comparing the fluxes generated through sonic#1 during the entire campaign period, the averaged theoretical damping corrections for all gases were

relatively larger than the experimental ones (see Figure S8), such as for $NH_3$ flux were 39% versus 28%, respectively. Surprisingly, even without extra distance separation between LI-7500DS and sonic#1, the TEO method still generated 12% correction for $CO_2$ flux and water vapor flux, while ICO method results suggested the needed correction was much smaller, only 2-3% on average for the entire period.

## 5    Discussion

In this experiment, we had the unique opportunity to use two newly developed instruments that share one essential feature needed to improve $NH_3$ flux measurements: neither of the instruments has an inlet line. These two instruments provided independent data for the flux estimate.

### 5.1    Field performance

The field performance of the miniDOAS and HT setup was assessed during this campaign. The miniDOAS system was steadily

housed in an air-conditioned container, and was operational and measuring close to 100% of the campaign period. In the campaign, 35% of this time was spent on intercalibration of the two miniDOAS instruments, and the rest on flux measurements. Towards the end of the campaign, after the HT had been removed from the field, the miniDOAS instruments were intentionally kept in cross-setting to confirm stability of the baseline of the concentration difference for two extra weeks. These intercalibration measurements confirmed the baseline stability of the $NH_3$-difference measurements was better than 0.002 µg

$m^{-3}$ drift over a seven-week period. In future applications, the frequency and duration of the inter-instrumental calibration of the two miniDOAS instruments can be further optimized, increasing the percentage of operational flux measurements to well above 65%. The miniDOAS optical system was almost insensitive to degradation, although the parabola mirror and the lamp may need replacement after about a year. Hence, we conclude that the miniDOAS gradient setup is field ready in its current configuration, also for longer-term measurements.


For the HT, ~ 21% missing data were caused by raindrop or dew on the optical mirrors, and the coating material of the mirror gradually deteriorated along time over the five-week period presumably due to rain as well. In addition, the HT instrument needed regular operator intervention (e.g. mirror cleaning). To make the instrument suitable for longer-term monitoring, in particular in areas with frequent rainfall, this needs to be addressed in future versions.

### 5.2    Flexibility in application

For operation in the field, the miniDOAS system requires at least about 10 meter, preferably 20 meter, of steady surface between container and retroreflectors. It also requires structural stability to maintain the alignment between the miniDOAS instruments and their retroreflectors. That is feasible for ground-level operation but more difficult for a site with tall vegetation, for example when evaluating deposition above forest canopies from a tower. Besides, the miniDOAS instruments also need ~

200 W at 230 V each, and are operated from a container ($2 \times 2 \times 2$ m) with air conditioning. Hence, its operation depends on a substantial mains power supply. The light-weight and portable HT instrument, on the other hand, currently only needs a 12 V, 50 W power supply permitting use at remote sites without access to mains power. It can be supported by a solar panel and a battery.



### 5.3 Concentration comparison

A substantial and varying discrepancy in $NH_3$ concentrations was found between the HT and the miniDOAS at similar measurement heights (average discrepancy -5.3, range -15 to + 6 µg m$^{-3}$). The miniDOAS instrument was initially designed for measurement of the absolute concentration level. It is currently used for concentration monitoring in the Netherlands and has a validated accuracy of better than 3%. Therefore, we concluded that the observed discrepancy was caused by a substantial and varying offset in the concentrations recorded by the HT, which correlated with the changing ambient air temperature

(Figure S4, $R^2 = 0.68$). In an earlier study testing the performance of the HT, Wang et al. (2021) compared measured $NH_3$ concentrations of the HT to those of a Picarro closed-path laser-based $NH_3$ analyser under steady lab conditions. The difference was within 10% during a 14-h experimental period. However, the indoor air temperature during that relatively short experiment would likely have been fairly stable, so any potential impact of temperature variation on concentration measurements by the HT could not easily be detected. In our field experiment, after applying the correction function to the HT concentrations using

the air temperature data, the agreement of concentration between the miniDOAS and HT was substantially improved. It remains to be seen if the parameters for this temperature correction function change over time and under different temperature ranges. Having the absolute concentration level right is especially important when using it to interpret the flux data beyond the net flux, and also when calculating deposition velocities. Since not all users have a separate, reliable concentration measurement alongside the HT in the variable field conditions, it is important to improve the accuracy of measured concentrations of the

HT itself. The effect may be caused by the external heat-exchange unit of the HT, which does not provide a temperature lock. To reduce the influence of variable temperatures on the absolute concentrations, a Peltier stabilised external cooler could be tested along with the HT.

### 5.4 Flux comparison

We evaluated the AGM and EC fluxes using standard, established analysis techniques for both. When the wind came from the

"green" sectors – where upwind terrain was relatively homogenous and obstacle-free, the overall pattern of the fluxes and the diurnal pattern agreed remarkably well between the DOAS-AGM and the HT-EC setups. Moreover, the fluxes showed a clear bidirectional behaviour switching between emission and deposition and also between day and night. Larger differences between the AGM and EC fluxes were observed for the other wind directions (Figure 10). The discrepancies between the AGM and EC fluxes can have several causes.

When measuring fluxes, the terrain upwind of the instruments ideally needs to be obstacle-free. Obstacles interfere with both measurement techniques, as they affect atmospheric turbulence pattern and disturb the $NH_3$ gradient. This condition was not met for all wind directions. In our case, all instruments were ~ 60 m away from the 213 m high tower especially when wind came from southeast (red wind sector). The fluxes were less comparable between AGM and EC than when fluxes originated from obstacle-free area (green wind sector), but the correlation was still modest ($r = 0.69$) as both measurement sets were

under the similar influence from such a big obstacle. When the wind blew from the north, obstacles in that direction were smaller, but still agreement between fluxes from AGM and EC was poorest ($r = 0.33$) and even occasionally showed fluxes in oppositive directions. This may have indicated that under such conditions, the heterogeneity of footprint area, for example caused by sheep grazing the research site, could have played a bigger role on measured fluxes by two systems.

Both the EC and AGM techniques assume spatial homogeneity of the surface-atmosphere fluxes and terrain within the

footprint. Due to the differences in measurement height and path- versus point-sampling, the AGM and EC setups 'see' different areas (Loubet et al., 2013). The footprint analysis for the EC method showed that measurements were representative of the terrain up to approximately 100 m upwind (Figure 13). Because of the lower measurement heights, the footprint of the AGM setup is expected to be smaller. In addition, the footprint of the upper and lower miniDOAS instruments was different. Thus, if either the terrain or fluxes were inhomogeneous, the AGM and EC setups may have captured different $NH_3$ fluxes. At

this measurement site, spatial homogeneity may partly be violated by the ditches in the terrain. Direct emissions may be





spatially inhomogeneous due to manure or fertiliser application or excreta from grazing animals. Fertilization and grazing also impact the nitrogen status of the vegetation. This leads to an enhancement of the nitrogen content of the vegetation, which could lead to stomatal emissions of $NH_3$ in daytime. To confirm the occurrence of potential stomatal emissions of $NH_3$, however, further interpretation of the measured $NH_3$ fluxes is needed. This is considered beyond the scope of this study but will be addressed in an upcoming paper.

We found that HT concentrations can differ substantially from miniDOAS concentrations. The differences were strongly linked to ambient temperature. In our current analysis, we treated these differences as a temperature-dependent offset. As ambient temperatures only changed gradually in time, so did the applied offset in the correction. As a consequence, this correction has virtually no impact on the HT flux measurement, as the flux measurement is based on observed concentration variations on a short timescale.

It is however not clear if the effect of temperature is limited to inducing just an offset in concentration. There could also be an influence on the span: at higher temperatures the HT might be more, or less, sensitive to $NH_3$. This would affect the flux measurement by the same factor and could be an explanation for the discrepancy in flux between miniDOAS and HT during daytime (Figure 11).

To eliminate this possible source of discrepancy, further studies to the cause and the exact effect of the temperature on the offset and slope of the HT calibration are necessary. The cause of the sensitivity to ambient temperature may be instrumental, or spectral.

In this paper, commonly-used flux calculations were used for both techniques. For the AGM method, standard calculations were used that are well established and has been used for over two decades now (e.g. Businger, 1986). For EC flux, the standard

analysis method in EddyPro was used, including the theoretical correction for high-frequency losses (TEO). Current instrument setups, however, are different from regular AGM and EC instruments (path versus point for miniDOAS, larger measurement volume for HT) and are applied to a new gas. These analysis techniques may therefore need adaptations. For example, we also tested an alternative experimental analysis method (ICO) to correct high-frequency losses for the EC and found different flux results.

The effect of the theoretical damping correction in EddyPro adds about 40% to the raw flux data. The estimate of Moncrieff et al. (1997) relies on the comparison of the theoretical frequency distribution of the $NH_3$ flux considering the damping through spectral transfer functions with a theoretical turbulence contribution per frequency. When using the empirical (ICO) method as described by Wintjen et al. (2020), using the cospectra for both the temperature and $NH_3$ turbulent data and matching these two the damping effect would only be 30% (about half of that due to sensor separation and half due to the sensor size). In that

case, the mismatch between the AGM and EC fluxes would increase to 50% instead of 30% (slope of the x-y fit in the green sectors in Figure 10and Figure S10). Since we do not have a very large dataset and because the empirical method can only properly run on the subset of the data that has fluxes large enough to make a reasonable spectral distribution, the fit of miniDOAS vs. HT shows more scatter (Figure S10). The benefit of the theoretical approach in EddyPro is that the theory is always available since it relies only on the arrangement and dimensions of the instrument. Therefore, we choose to present the

comparison based on the EddyPro calculation but strongly advise further evaluation of the damping calculation method. Similar advice is given for the AGM method, where almost everyone uses the standard stability correction functions and these bring a generalisation that might be not fully representative at a given measurement site. Hence, it is not surprising to find that when comparing the ICO method $NH_3$ flux to the theoretical AGM method, the correlation is worse.

Part of that damping was related to the sensor separation, which was outlined above. Reducing the distance between the sonic

anemometer and gas analyser to closer than the 40 cm heart-to heart distance we used is not recommended, because then the instruments can be too close to touching each other. The damping correction might get smaller but the HT instrument is likely to distort the airflow due to shadowing effects of the sonic transducers (Horst et al., 2016).



### 5.5 Outlook

For the Netherlands, the main goal for these measurement sets is to evaluate the deposition levels of $NH_3$ in Natura 2000 areas
(European Union, 1992). The flux levels in these areas are generally low and less complex in temporal structure. Little or no
(re-)emission is to be expected. To further test the performance of the flux instruments for these conditions, the comparison of
the miniDOAS and the HT instrument should be continued at more homogeneous sites. Attention should be given to avoid
nearby obstacles and nearby animal emission sources within the footprints. These intercomparisons could also include further
studies into the temperature sensitivity of the HT, and tests for remedies thereof in future versions.

The stability of the miniDOAS instruments need to be confirmed for longer-term monitoring applications. In particular, the
duration and frequency of the intercalibration periods needs to be optimized to allow a larger fraction of the measurement time
in flux-mode.

In a follow-up study, the effect of different flux data processing algorithms can be further quantified. These studies need to
address the different processing options for both techniques and their effect on the calculated fluxes. For consistency, we
suggest processing different techniques on a similar level of complexity, i.e., representing the local atmosphere on the same
level of detail and using either theoretical or site-specific corrections. For example, if actual, measured turbulence spectra are
used in the EC-analysis, locally-derived flux-profile relationships should be used in the AGM-analysis.

### 6 Conclusions

We compared two novel open-path optical instruments to measure $NH_3$ concentration and flux during a 5-week comparison
period at Cabauw, the Netherlands: two active custom-designed broadband UV-based miniDOAS (Differential Optical
Absorption Spectroscopy) instruments and a commercially available infrared-based quantum cascade laser HT8700E gas
analyser developed by the company Healthy Photon (HT). Both instruments avoid the hysteresis effects caused by the
stickiness of $NH_3$ to tubing and instrument interior, and are as such insensitive to interference by ammonium aerosols. Both
instruments showed good uptime during the campaign. The uptime of the miniDOAS system reached 100% once operational,
but regular intercalibration of the two instruments was applied to test baseline stability (35% of the 7-week uptime).
Intercalibration time can be reduced in future application based on the results of this campaign. The HT does not measure
during rain, or shortly after rain while the instrument is drying. The coating of HT mirrors tended to degrade because of rain,
causing 21% data loss during the 5-week uptime.

The miniDOAS system measured fluxes using the aerodynamic gradient method (AGM), the HT8700E measured fluxes using
the eddy covariance (EC) method. After data quality filtering, a total of 848 simultaneous half-hourly flux measurements were
compared, showing that both instruments gave similar values for the $NH_3$ exchange ranging from *ca.* -80 to
140 ng $NH_3$ m$^{-2}$ s$^{-1}$ (Figure 10). When the upwind terrain was both homogenous and free of nearby obstacles within around
100 m, the two systems showed the strongest correlation (n = 113, *r* = 0.87) and provided similar temporal patterns. In addition,
780   the observed diurnal pattern of the two systems had the same shape (Figure 11). As such, the deposition flux during night-time
was ca. 25 ng $NH_3$ m$^{-2}$ s$^{-1}$ (equivalent to 465 mol $NH_3$ ha$^{-1}$ yr$^{-1}$). The highest emission occurred around noon and was up to 50
ng $NH_3$ m$^{-2}$ s$^{-1}$. Moreover, the AGM flux values were larger than the EC ones during daytime.

The uncertainty analysis showed that the random error of the two systems was similar (Figure 12). The median relative random
785   errors were 23% for the AGM flux versus 15% for the EC flux. The mean random error (1σ) for half-hourly flux values of the
miniDOAS was about 9.8 ng $NH_3$ m$^{-2}$ s$^{-1}$, and its maximum value did not exceed 15 ng m$^{-2}$ s$^{-1}$. For the HT, the mean and
maximum random errors were 5.5 and 10 ng $NH_3$ m$^{-2}$ s$^{-1}$, respectively. These values are adequate to allow the study of
deposition and emission processes. The random errors of both techniques varied substantially with meteorological conditions





and time-of-day. For AGM flux, it was relatively higher during daytime. The diurnal cycle in the random error of the EC was, on the other hand, far less distinct.

While flux measurements between HT and miniDOAS in general compared well, we found a substantial variable offset in the HT concentrations. They were sensitive to air temperature, causing substantial differences (range: -15 to + 6 µg m$^{-3}$) between the two systems. In this study, we used the miniDOAS as a reference to correct the HT concentration using a temperature-dependent offset and assuming no impact on the span. It should be stressed that these offset corrections only have an impact on the HT concentrations, not (or only very minor) on the HT fluxes. However, a temperature dependency in the span would also affect the HT fluxes. Further studies into the temperature dependence of the HT concentrations are needed to confirm the span calibration is indeed not impacted by changes in temperature.

The footprint analysis for the EC method showed that measurements were representative of the terrain up to approximately 100 m upwind. In the southeast direction, the footprint size was much smaller due to the meteorological measurement tower, which largely blocked the air flow. The footprint size of the AGM was not analysed but is expected to have a similar shape. Moreover, because of the lower measurement heights, the miniDOAS system is expected to have a smaller footprint, and the footprints of upper and lower paths are substantially different.

Spatial heterogeneous flux patterns need to be avoided in the upwind footprint region as they can influence the result and render interpretation more complicated or even impossible. Also, the 10% difference found between the theoretical (Moncrieff et al., 1997) and empirical (Wintjen et al., 2020) method for correcting high-frequency losses of EC fluxes may be related to inhomogeneities in the footprint area since they were not reproduced by theoretical cospectra. In addition, the terrain within all footprints needs to be homogeneous in its vegetation type and roughness. For further intercomparisons, obstacles-free, cattle-free, more homogeneous surroundings are highly recommended.

In deposition studies and parametrisations, reliable concentration and flux values are both needed. The miniDOAS provides both values reliably and appeared ready for long-term hands-off monitoring. The HT is presented solely as a flux instrument, and makes no claim to being an accurate monitor for NH$_3$ concentrations yet. In addition, the current system had a limited stand-alone operational time under the prevailing weather conditions.

In this study, we demonstrated that the miniDOAS and HT8700 systems provide comparable flux measurements at half-hourly time resolution. Under the right circumstances, data from both instruments can facilitate the study of processes behind dry deposition in different ecosystems, allowing better understanding and better parametrization of these processes in chemical transport models. These observations also enable to test and validate low-cost deposition measurement systems like the conditional time-averaged gradient (COTAG; Famulari et al., 2010), or inferential deposition networks (e.g. those listed by Walker et al., 2020).

## 7 Author contribution

DS, AH, SR and TvG designed the study. DS and AH coordinated the field campaign. DS, SB, RvdH and MH developed and tested the miniDOAS instruments and performed measurements during the campaign. JZ, AH, AF and PvdB performed the EC HT measurements. JZ and PW processed the EC and HT data and determined the EC NH$_3$ flux. SB processed miniDOAS concentration data, and SvdG determined the AGM NH$_3$ flux. Figures were made by SvdG, SB and JZ. DS, AH and SR prepared the manuscript with contributions from JZ, SvdG, and SB. RS, PW and TvG reviewed and corrected the draft manuscript.



**8    Competing interests**

The authors declare that they have no conflict of interest.

**9    Acknowledgements**

This work was done partly in the framework of the Dutch Ruisdael program (https://ruisdael-observatory.nl) and was part of the annual Ruisdael campaign (RITA-2021). Funding from the Ministry of Agriculture, Nature and Food Quality (LNV) is gratefully acknowledged. We thank the Royal Netherlands Meteorological Institute (KNMI) for site access and assistance during the campaign, especially Arnoud Apituley for coordination and help with site selection. We thank Dr. Kai Wang, from Institute of Atmospheric Physics, Chinese Academy of Sciences, Beijing; and staff from Healthy Photon Lt. Co, especially Dr. Yin Wang and Dr. Peng Kang are for helpful discussions about data processing of the HT. Furthermore, we thank Daniëlle van Dinther (TNO) for merging various data streams during the campaign. We acknowledge RIVM colleagues Kim Vendel for the preliminary processing of the AGM data at beginning of the campaign and Miranda Braam for helping with the uncertainty analysis.

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
