# Peer review of "Measuring dry deposition of ammonia using flux-gradient and eddy covariance methods with two novel open-path instruments"

_Atmospheric Measurement Techniques, 2022_

## Referee Comment (RC1)

**General comments**

This paper is reporting a comparison of the measurement techniques, flux gradient and eddy covariance, in measuring $NH_3$ deposition fluxes. The benefits of this study would be great for the gas measurement community, especially for the peoples who are interested in $NH_3$ emissions and deposition because the nature of $NH_3$ "sticky" character and having practical friendly large-scale field equipment make the filed measurements difficult. However, there are some drawbacks in the current stage of the manuscript, I recommend authors to address the issues before considering to be published in this journal.

In the abstract, I would expect to see some clear messages including the measured $NH_3$ deposition fluxes measured by both techniques and the difference in the $NH_3$ deposition fluxes between these two techniques, especially during the periods when winds came from no-objects direction (green sector). Secondly, in the methods and materials section, there is lack of information on the soil physical and chemical properties along the footprint or the transects from 100-200 m away to the tower, particularly within the green wind sector. The soil properties, including N and C contents, moisture contents, pH, soil texture, and grass types, crop age etc, could help us better understand the upwind footprint areas and how they could contribute to the measurements in this study. Thirdly, in the results and discussion sections, authors spent a length to discuss the footprint. My main concern is that the two techniques have different footprints due to the height of the sensors, and different footprint area could contribute to the different $NH_3$ emissions and deposition (due to the land use and farm management practices). EC is often used at a larger area and but can't be deployed at many filed studies due to the limited size of the paddocks, which requires a difficult "footprint" analysis. I suspect there is lack of an accurate footprint modelling to correct EC measurements in this study. see some studies from Coates et al., 2018, 2021.
*Coates, T. W., Benvenutti, M. A., Flesch, T. K., Charmley, E., McGinn, S. M., and Chen, D.: Applicability of Eddy Covariance to Estimate Methane Emissions from Grazing Cattle, J Environ Qual., 47, 54-61, 10.2134/jeq2017.02.0084, 2018.*

Reviewer suggests that there is a need of adding a footprint analysis of AGM measurements as well. In addition to the factor of u*, we may also want to look at the correlation between AGM footprint and stability length L, which also can tell us the footprint variations during night-time stable and day-time unstable conditions. Further information on different paddocks soil properties and the history of farm management in the last couple of weeks (grazing, fertiliser application, irrigation, ploughing etc) are needed. Majority of N losses as $NH_3$ occurred at the first 2-3 weeks following N fertiliser application to the soils.

**Specific comments**
1, be aware of self-citation. There are too many times using two references Wang et al., 2021, 2022. I'm sure there are many other studies in this area.

2, the manuscript is long. Please remove some repeated parts and shorten unnecessary contents, for example, line 134 to 137.

3, add a detail map of experimental site including the surrounding terrains (200 m radius) and indicates equipment locations, heights, and the dimension of each paddock.

Line 37. What does "right circumstance" mean here.

Figure 2. perhaps need number each instrument.

Line 212. please explain why the path-length was set at 22.1 m (between miniDOAS and retro-reflector). Was it recommended by the manufactory? Is there any specific reason that the distance between two miniDOSA paths (upper and lower path) was 1.53 m? would the measurements be better if the distance between the two paths is larger, such as 2.5 m?

Line 290 please provide the details of the functions for stable and unstable conditions.

Line 307. It seems to me that the signal to noise ratio corresponds to the detection limit of $NH_3$ flux. What caused the signal to noise ratio higher in the study by Wang et al., 2022?

Line 311. was there any drifting of the release rate during the measurement period? if yes, what was it?

Line 321-322. It is a concern that weather the HT8700 sensor can be used in a wet season as more $NH_3$ deposition will be expected in wet season than dry season.

Line 372. Provide the value for A, B, and C parameters. Are these values the same as that reported in Wang et al., 2021? Would these values be consistent or variable at different environmental conditions?

Line 379. More details about the stationarity and integral turbulence tests used in this study are needed.

Figure 8. Many studies have shown that NH3 emissions positively correlated with ambient temperature, higher emissions in the middle of the day (higher ambient temperature) than that at night-time, however from the figure 8, both miniDOAS bottom and miniDOAStop show the opposite pattern, higher concentrations at low ambient temperature and lower ones at higher temperature. Why? Perhaps add one or two sentences to mention that at night-time with low wind conditions, the concentrations can be build up and higher concentrations at night-time than day-time were observed. Suggest adding wind speed (or u*) in the figure.

Line 524. I don't agree, as the better agreement was between 21-24 Sep, while much higher AGM before 18 Sep. Again, suggest comparing the concurrent fluxes from both techniques. Furthermore, should consider using the footprint model to correct the EC fluxes.

Line 531. Is it necessary to calculate the cumulative flux on this particulate day when the winds were from the SE, large disturbance from the objects?

Figure 11. 1) why is this typical NH3 diurnal pattern different from the NH3 concentration plotted in Figure 8?

2) From Figure 9, the top panel shows most of time EC measurements were higher than AGM, but in lower panel it shows some time AGM higher than EC, other time EC higher than AGM, and some time they agreed well. However, in Figure 11, obviously AGM (in blue) is higher than EC (in red). Why?

It is import for science aspect, it is worth to split the results, to identify in which conditions EC higher than the red and when AGM higher than EC.

3) please indicate if the same numbers listed on the top present for both technique measurements? If not, please add different numbers.

4) there is lack of explanation on the diurnal pattern. There is a "jump" in the deposition flux in the morning at 5 am, was it really happening or just due to the noise? There are not many available data at night for deposition flux datasets (both only 2 at 5 am and 22 pm).

Line 595. I would like to see how the footprint are associated with the surface roughness z0 and stability length L (stable and unstable conditions).

Line 660 please indicate it in the Figure 1.

Line 802. Please provide a footprint analysis for AGM, the EC measurement should be corrected with a footprint modelling.

**Technical comments**
Line 410. Remove "respectively"
Line 413. Add a comma, to be 5%, 5%, and 1%, respectively.

---

## Referee Comment (RC2)

First I would openly state that I have a long lasting collaboration with some of the authors and I closely followed and discussed their progress. I cannot exclude that I am biased in this sense.

This paper presents an NH3 flux intercomparison with two novel open-path instruments. The major advantage of open-path instruments is the absence of an inlet line that is notoriously influencing high frequency concentration measurements needed to determine fluxes. The measurements took place at the Cabauw station located in the middle of the Netherland. It is an open landscape  mostly used as intensive grassland. From a micrometeorological perspective it is good location as most of the time good turbulent conditions are present.

I am very pleased to see the great progress that was achieved regarding the accuracy and precision of the DOAS system. The systems have no reached a level that meaningful  vertical gradient measurements are possible. Also the open path system from Healthy photons is precise and fast enough to perform reliable EC measurements.

Looking from a greater distance the overall results summarized with figure 11 are looking very nice. Both systems show similar NH3 fluxes and more important they show the same diurnal structure with identical changeover from deposition to emission. So it is very likely that both instruments do measure an existing mean flux over the surface determining the flux.

I could stop here and say publish as is with some minor corrections already addressed by the first reviewer. But this is not my understanding of a serious review and I will dig in the following a little bit deeper. But keep in mind that this is complaining at a high level.

Specific issues:

I think that the title is not appropriate. The work focus on an intercomparison of two approach to determine bidirectional NH3 exchange. The title should reflect this.

It would be important to describe shortly the "flux landscape" of the chosen site. It is located in a an area with intensive animal production associated with a high NH3 turnover. Generally small areas with high emission densities (stable, storage, manure fields) are surrounded with intensively managed field that are expected to show all the time changes from emissions to deposition mainly controlled by their actual N-status (see e.g. the year around measurements that we did at the Oensingen grassland site many years ago, Flechard et al., 2010 https://bg.copernicus.org/articles/7/537/2010/). Judging from a google maps picture from the area there are many small individual fields as well as stationary sources within the footprint area.

[Figure]

Deposition mainly occurs during night stable conditions when NH3 is concentrated in the boundary layer and emission occurs during instable conditions when the growing boundary layer leads to NH3 concentration below the "system compensation point". Consequently an inverse relation between concentration and fluxes must be expected, even though at a first glance this seems contra intuitive. In this context I was very puzzled by the sentence in line 488 *"The highest concentrations were observed by both systems at noon when air temperature reached the highest level of the day (Figure 8)".*

I am uncertain whether this is just a slip of the pen or a misunderstanding.

The evaluation of the EC data is state of the art. The most important correction is the high frequency damping in the order of 20 to 30%. I am aware that the empirical ogive method needs a lot of good data. I suggest to use only flux data with a good looking covariance function for both NH3 and temperature flux and not using daily mean values.

DOAS gradient measurements require a demanding procedure to be precise enough that includes measurements in cross position. I rate this as a very positive development of the DOAS approach that I was not expecting based on our own experiences. But I see a tendency to overestimate the precision of the DOAS measurements. E.g. in line 280 it is stated *"that differences can be measured well below our target precision of 0.1 µgm-3."* In line 449 it is stated *"The random error of the miniDOAS NH3 concentration differences … was determined to be 0.088 µgm-3".* This is below the target limit but not well below.

I also have an other interpretation of the cross position data. In the second period I judge that there is a systematic offset in the order of 0.14 µgm-3 as this is a consistent value for good wind conditions. In case the wind is coming from the red sector I don't see a mechanism that could produce the measured difference between the two crossed paths, especially for the lower concentration range.

The obstacle upwind will affect mostly the turbulence and not the NH3 concentration. You could also check the EC-NH3 covariance function and time series to see whether there is a major disturbance.

As far as I know there are many influences that potentially influences concentration determination in the order of 0.1 to 0.2 μgm-3 especially under field operation. Judged from figure S5 such a correction could at least partly explain the higher NH3 emission of the AGM approach around noon.

The whole footprint discussion should be modified. In case of the EC approach the calculated fluxes are representative for a small volume at a height of 2.8m. In case of the AGM approach it is a mean vertical flux integrated over a 22m path between 0.76 and 2.26m assuming that the used transfer velocities correctly reflect the atmospheric turbulence. These fluxes need to be translated into exchange fluxes at the soil surface. The simplest approach and implicitly used in most cases is that the vertical flux is constant in all 3 dimension and consequently the measured fluxes are equal to the surface exchange flux. In such conditions footprint considerations are not an issue.

The footprint climatology shown in Figure 13 gives a good impression which area will on average determine the measured fluxes. But the analysis should be adapted to the selected data point (green and light green conditions).

NH3 exchange fluxes will most likely be different between all the small fields in the neighborhood of the measuring point within the footprint. Consequently EC and AGM approach will considerably differ as the footprint density function for the two approaches differ. Note also that this function will considerably differ for the lower and upper DOAS path.

I also suggest to separately calculated the footprint density function for stable and instable conditions. By the way the bls-R tool developed by Christoph Häni is appropriate to do so. It allows to calculate concentration and flux footprints μgm-3 (https://github.com/ChHaeni/bLSmodelR). It would interesting to overlay these footprint density function on a land use map. This would allow a plausibility control whether the recorded differences in Figure 11 reflect a topographical effect or whether systematic effects in the measurement systems are in the focus. But an in depth footprint analysis needs detailed information on the land use.

To summarize I suggest to do:

- Reconsider the title
- Add a paragraph regarding the expected flux landscape at Cabauw. This can be integrated to the discussion of the concentration characteristics.
- Down scale the footprint issue. The presented fluxes assume a homogeneous vertical flux and that the used set of turbulence parameter reflect the present meteorological conditions. A consideration of the footprint is for another paper.

For my curiosity I would like to see in a supplement or in the main paper a plot with the AGM concentration differences, the transfer velocities and the fluxes (extension of figure S5). Regarding the EC data it would be helpful to see NH3'w' covariance function as well as ogives for a few cases , both from the green and red sectors. (not only the best!).

---

## Author Comment (AC1)

**Response to Reviewer 1**

**General**

We thank both reviewers for their thorough reviews and their wealth of suggestions. We greatly appreciate their input. We recognize in their reactions that the progress that we have made in the described research now opens up a large number of new questions: our research team felt exactly the same. With the reviewer's help we now harvested a number of good ideas to further optimize the data evaluation of this experiment, but more important: of experiments to come. Because facing the editor's request to reduce the length of the paper substantially does not allow expanding the evaluation on all aspects highlighted by the reviewers. We have tried to find a balance between which parts of the review comments we take on board now, and which parts we will take with us to our next campaign.

A main decision is how to deal with the footprint issues raised when evaluating the data. In the past, the Cabauw site has worked very well as a location for many flux intercomparison experiments with other gases (e.g. Peltola et al., 2014 https://bg.copernicus.org/articles/11/3163/2014/). However, this campaign has shown that it is probably somewhat too inhomogeneous for detailed intercomparison of ammonia fluxes. This was aggravated by the wind directions during most of the campaign, which were different from the normally prevailing southwesterly winds. Considering that (1) we did not collect the detail of activity data in the direct surroundings for a proper evaluation of the impact of different footprints on the fluxes of EC & AGM, and (2) that we are urged by the editor to reduce the paper length considerably, we have decided to limit our analysis in this paper to the standard, 3D-homogeneous flux approach. We are aware that with this approach some of the observed differences will originate from footprint issues. These issues will get more emphasis and attention in our next campaign, that will be located in a more homogeneous area.

In the remainder of this document we will follow the reviewers text, and our pointwise reactions and answers can be found in *italic blue text* at the applicable position.

On behalf of the author team,

Daan Swart.

**General comments**

This paper is reporting a comparison of the measurement techniques, flux gradient and eddy covariance, in measuring NH3 deposition fluxes. The benefits of this study would be great for the gas measurement community, especially for the peoples who are interested in NH3 emissions and deposition because the nature of NH3 "sticky" character and having practical friendly large-scale field equipment make the filed measurements difficult. However, there are some drawbacks in the current stage of the manuscript, I recommend authors to address the issues before considering to be published in this journal.

(1) In the abstract, I would expect to see some clear messages including the measured NH3 deposition fluxes measured by both techniques and the difference in the NH3 deposition fluxes between these two techniques, especially during the periods when winds came from no-objects direction (green sector).

*Our primary aim of the campaign was to test if both novel instruments were indeed capable of measuring the dry exchange flux of ammonia at high temporal resolution, to determine under which conditions, and for how long. Instrumental errors were found and corrected, and a large part of the data analysis was developed further during the campaign and its analysis afterwards. A thorough intercomparison of the results, focusing on the observed differences between the measurements of both instruments and the underlying causes is a large step further, would require considerably more effort, and was outside our scope for this first campaign. Actually, this first campaign was needed to demonstrate that such a study would now be feasible and meaningful, and to give us -at least to some extend- the clues on how to set it up and what additional information would be needed.*

*However, we agree with the reviewer that where possible additional indicative numbers are relevant to illustrate the flux comparison results. Hence, we have given those comparative numbers between line 25 and 27. To further clarify the two methods' differences in the green sector, we will provide an extra sentence in the abstract following Line 27, related to the green sector net flux:*

*"While differences in cumulative fluxes were small (~10%) as long as the upwind terrain was homogeneous and free of nearby obstacles."*

(2) Secondly, in the methods and materials section, there is lack of information on the soil physical and chemical properties along the footprint or the transects from 100-200 m away to the tower, particularly within the green wind sector. The soil properties, including N and C contents, moisture contents, pH, soil texture, and grass types, crop age etc, could help us better understand the upwind footprint areas and how they could contribute to the measurements in this study.

*We agree that such information would be needed to better understand the measurement results, and especially the differences between the techniques, given the fact that the effective footprints of both instruments are different. However, as stated above, this was not our primary aim for this campaign, and for this campaign we did not gather the information needed for this analysis. For this paper, we limit our analysis, assuming a 3D-homogeneous flux field and no footprint issues (knowing that this is not completely in line with reality). In fact, we think that the Cabauw site may be too complex an environment for such a more elaborate analysis. We would like to address this issue first in a simpler environment in our next campaign.*

[Figure]

*Figure 1. The area surrounding the Cabauw measurement site. Cabauw is in a flat area at -1m, being in the delta of the river Lek shown in the south east. The line with housing going east-west, running north of the tower has a series of farms. Map from [www.pdok.nl/](www.pdok.nl/) (downloaded 07-02-2021).*

*To illustrate the different land cover classes, we will also add to the Supplementary Materials a simple land cover classification based on a Google Map (RGB) image, derived using Flux Footprint Predictions (https://geography.swansea.ac.uk/nkljun/ffp/www/). The Flux Footprint Predictions tool derives an unsupervised land cover classification based on a Bing or a Google (RGB) map for the footprint area and overlays the footprint climatology with the classification, providing a simple estimate of what land cover contributes most to the measured fluxes. The unsupervised classification will derive five land cover classes. This is only a very simple approach for land cover classification that cannot substitute a detailed analysis.*

[Figure]

*Figure 2. Five land cover classes and their distribution ratios under the EC footprint area, and the overall land cover classes. Percentages of each land class are shown in the individual land class diagrams. A comparison of the Google Earth image with the land cover diagrams suggests the following classification: Land class 5: wet grassland, in light blue (count 42.3%); Land class 2: less wet grassland, in orange (count 38.2%);Land class 4: hay land (harvested grassland), in bright blue (count 9.9%);Land class 1: ditches and drainage lines, in light yellow (count 8.2%);Land class 3: concrete road surface, in dark blue (count 1.4%);Land class 5 and 2 have a similar roughness height, while land class 4 has lower roughness.*

*The plot below also gives an illustration of the difference in management of individual paddocks in the flux footprint through time.*

[Figure]

*Figure 3. (c) land cover within a 300-meter radius in June 2021 (google earth image); (d) idem in March 2022.*

(3) Thirdly, in the results and discussion sections, authors spent a length to discuss the footprint. My main concern is that the two techniques have different footprints due to the height of the sensors, and different footprint area could contribute to the different $NH_3$ emissions and deposition (due to the land use and farm management practices). EC is often used at a larger area and but can't be deployed at many filed studies due to the limited size of the paddocks, which requires a difficult "footprint" analysis. I suspect there is lack of an accurate footprint modelling to correct EC measurements in this study. see some studies from Coates et al., 2018, 2021. *Coates, T. W., Benvenutti, M. A., Flesch, T. K., Charmley, E., McGinn, S. M., and Chen, D.: Applicability of Eddy Covariance to Estimate Methane Emissions from Grazing Cattle, J Environ Qual., 47, 54-61, 10.2134/jeq2017.02.0084, 2018.*

*We agree with the reviewer that the analysis of the EC and AGM techniques becomes more difficult if the terrain is complex in spatial structure and land use varies with time. We also agree that the two instruments have different footprints, which also vary in time. This will definitely be a source of differences between the results of the two instruments. In this study, we analyse our data assuming a 3D-homogeneous flux field, as is common practice in many flux studies, and -we feel- a good approach given the aim of our campaign. A next campaign focusing on a thorough intercomparison of the techniques can best be undertaken at a more homogeneous, simpler terrain without local emissions. We feel the Cabauw location is too complex for that.*

(4) Reviewer suggests that there is a need of adding a footprint analysis of AGM measurements as well. In addition to the factor of u*, we may also want to look at the correlation between AGM footprint and stability length L, which also can tell us the footprint variations during night-time stable and day-time unstable conditions.

*We agree that further work on the footprint analysis is needed for both instruments, and especially for the AGM (miniDOAS), as here we have data taken over two paths (not points) and at two different heights. As stated above, in the current study we assume a spatially homogeneous flux field, so in this analysis the footprint issues may be ignored (but not forgotten).*

(5) Further information on different paddocks soil properties and the history of farm management in the last couple of weeks (grazing, fertiliser application, irrigation, ploughing etc) are needed. Majority of N losses as $NH_3$ occurred at the first 2-3 weeks following N fertiliser application to the soils.

*See our earlier remarks under (1) and (2).*

**Specific comments**

(6) 1, be aware of self-citation. There are too many times using two references Wang et al., 2021, 2022. I'm sure there are many other studies in this area.

*The HT is a relatively new, commercially available instrument, and the Wang et al (2021, 2022) papers are currently the only peer-reviewed sources relating to this instrument. As testing the HT instrument is a core subject of this paper, we refer to these publications regularly, when relevant.*

*In addition, it should be noted that Wang et al. are not co-authors of our paper. Nor are we co-authors in their paper. So this is not a matter of self-citation.*

(7) 2, the manuscript is long. Please remove some repeated parts and shorten unnecessary contents, for example, line 134 to 137.

*The editor and the other reviewer have also indicated that the paper needs to be shortened. We aim to reduce the size by eliminating unnecessary information and repetition, and by moving the footprint analysis largely to the supplemental material.*

(8) 3, add a detail map of experimental site including the surrounding terrains (200 m radius) and indicates equipment locations, heights, and the dimension of each paddock.

*In the "Campaign setup and Site" section, the combination of the current Figure 1, with the addition of the figure mentioned in our answer under point (2) under cover both the locations of the instruments and the surrounding area.*

(9) Line 37. What does "right circumstance" mean here.

*We will adapt the text to: "under relatively dry, low-dust conditions".*

(10) Figure 2. perhaps need number each instrument.

*We have tried this, but the photo is already quite busy and we feel adding these numbers did not help to get a clearer overview of what is what.*

(11) Line 212. please explain why the path-length was set at 22.1 m (between miniDOAS and retro- reflector). Was it recommended by the manufactory? Is there any specific reason that

the distance between two miniDOAS paths (upper and lower path) was 1.53 m? would the measurements be better if the distance between the two paths is larger, such as 2.5 m?

*In the paper, we refer to (Berkhout et al., 2017; https://amt.copernicus.org/articles/10/4099/2017/) and (Volten et al., 2012; https://www.doi.org/10.5194/amt-5-413-2012) for a description and more in-depth info on the miniDOAS. The answers to the reviewer's questions are: 22 meter provides a proper balance between (1) loss of light (mainly by Rayleigh extinction, and geometrically) over the 2 \*22 meter path, and (2) sensitivity to ammonia (as determined by the amount of differential absorption in the fingerprint absorption lines). A shorter path would have more light but be less sensitive, a longer path would have less light but be more sensitive. Both parameters together determine the precision of the ammonia measurement. Path lengths between about 10 and 25 meters can be used in practice. In this case 22 meters was chosen as this is the default separation in the Dutch monitoring network, and fits best to our calibration facilities. The bottom path needs to be low, but well above the vegetation. The top path should be as high up as practically possible, as more vertical distance leads to better sensitivity for the flux-induced gradient. In our case the practical limit was the height of the measurement container.*

(12) Line 290 please provide the details of the functions for stable and unstable conditions.

*The stability correction functions are described in the references given on the AGM theory. As mentioned in the manuscript, we use Paulson (1970; https://journals.ametsoc.org/view/journals/apme/9/6/1520-0450_1970_009_0857_tmrows_2_0_co_2.xml) for unstable conditions, and Beljaars and Holtslag (1991; https://www.doi.org/10.1175/1520-0450(1991)030<0327:Fpolsf>2.0.Co;2) for stable conditions.*

(13) Line 307. It seems to me that the signal to noise ratio corresponds to the detection limit of NH3 flux. What caused the signal to noise ratio higher in the study by Wang et al., 2022?

*There is a change from 0.30 ± 0.05 ppbv in the first study, to 0.41 ± 0.06 ppbv in the second study. We cannot speak for Wang et al., but to us these figures do not appear too different, for different campaigns under different conditions.*

(14) Line 311. was there any drifting of the release rate during the measurement period? if yes, what was it?

*We consider the way the factory calibration was performed to be outside the scope of this paper. It was not performed by us, so we do not know the finer details. We will remove this part altogether and instead refer to their paper for this.*

(15) Line 321-322. It is a concern that weather the HT8700 sensor can be used in a wet season as more NH3 deposition will be expected in wet season than dry season.

*We agree. Therefore we provided our feedback on this undesired limitation to the manufacturer. They are currently working to adapt the HT instrument such that operability under more humid conditions improves. We hope to test their next version soon.*

(16) Line 372. Provide the value for A, B, and C parameters. Are these values the same as that reported in Wang et al., 2021? Would these values be consistent or variable at different environmental conditions?

*A, B,C are variables that change with air temperature, pressure and water vapour density. Values change every half hour. The procedure for this is described in Wang et al, 2021)*

(17) Line 379. More details about the stationarity and integral turbulence tests used in this study are needed.

*In the text, we refer to Mauder and Foken (2006; https://doi.org/10.1127/0941-2948/2006/0167). The procedure is standard in EddyPro.*

(18) Figure 8. Many studies have shown that NH3 emissions positively correlated with ambient temperature, higher emissions in the middle of the day (higher ambient temperature) than that at night-time, however from the figure 8, both miniDOAS bottom and miniDOAStop show the opposite pattern, higher concentrations at low ambient temperature and lower ones at higher temperature. Why? Perhaps add one or two sentences to mention that at night-time with low wind conditions, the concentrations can be build up and higher concentrations at night-time than day-time were observed. Suggest adding wind speed (or u*) in the figure.

*We will correct an obvious error in line 488: the highest concentrations are observed during the night, not at noon. We will add some lines to explain the observed behaviour as requested. It should be well noted that higher **emissions** during daytime do not necessarily imply higher **concentrations** during daytime, as the mixing volume is completely different. See also our reply on (21).*

(19) Line 524. I don't agree, as the better agreement was between 21-24 Sep, while much higher AGM before 18 Sep. Again, suggest comparing the concurrent fluxes from both techniques. Furthermore, should consider using the footprint model to correct the EC fluxes.

*Effects of the spreading of manure are to be expected at least several days after the application, so if manuring stops at September 15, effects may be visible up till September 18 or even 20. That is, assuming all farmers obey the rules strictly. The green periods after September 20 (effectively 4 days) compare very well. We will change the line to clarify the exact period intended to: "In the green and light-green wind directions the NH3 fluxes from the two methods compared very well after September 20. In this period little or no effects of manure application should be present."*

(20) Line 531. Is it necessary to calculate the cumulative flux on this particulate day when the winds were from the SE, large disturbance from the objects?

*Figure S6 includes cumulative flux data only from the green and light-green wind sectors. So no winds from the SE are involved. We agree it may be unclear what we want to convey with this sentence. We will change the paragraph to:*

*"Considering only high-quality measured fluxes during this period, the cumulative daily fluxes of the AGM and EC were in general similar, with typical differences around 10%. When looking at the cumulative flux over the full period however, a larger difference is observed. This difference appears stepwise on a single day, September 24th. On this day, and only*

*during a few hours around noon, we see a much larger flux observed by EC compared to AGM. Most likely, the discrepancy is caused by footprint issues in combination with very local emissions. Unfortunately, we lack the means to validate this assumption."*

(21) Figure 11.

    a.   1) Figure 11. why is this typical NH3 diurnal pattern different from the NH3 concentration plotted in Figure 8.

*Flux and concentration are different quantities. During daytime, turbulent mixing is stronger and the boundary layer rises. Alle surface emissions are diluted and spread over a larger volume, resulting in lower near-surface concentrations during daytime. On the other hand, emissions during daytime increase because (1) temperature is higher and (2) the vegetation can more easily release ammonia when outside concentrations drop. This results in a higher surface flux during daytime.*

2) From Figure 9, the top panel shows most of time EC measurements were higher than AGM, but in lower panel it shows some time AGM higher than EC, other time EC higher than AGM, and some time they agreed well. However, in Figure 11, obviously AGM (in blue) is higher than EC (in red). Why? It is import for science aspect, it is worth to split the results, to identify in which conditions EC higher than the red and when AGM higher than EC.

*The reviewer poses an interesting question here. If both techniques are properly calibrated, one would expect random variation in which instrument gives the largest measurement result. If on the contrary the occurrence of e.g. AGM>EC occurs more often related to a specific outside parameter or condition, this definitely points to a need for further study.*

*The fact that we see AGM to be systematically larger than EC around noon in figure 11 and figure S7 is such a pointer. Yet, it is not a conclusion but just a starting point for further study. The origin may be related to temperature, solar radiation, wind speed, wind direction, but also to instrument-related parameters in one or both instruments, like instrument temperature, straylight, or footprint issues.*

*We already identified that local manure application is a probable partial cause of the observed discrepancies, as the differences in fig S7 are smaller than in fig 11. But the mechanism of that needs further research. It is currently unclear why these observations are different.*

*To identify different causes and split them from each other, a much larger dataset of paired observations would be called for, combined with the observation of external parameters as indicated above, including the soil conditions and agricultural management of the footprint area and direct surroundings. Therefore, this calls for a longer intercomparison in a simpler environment. This is one of the aims in our next campaign.*

    b.   3) Figure 11. please indicate if the same numbers listed on the top present for both technique measurements? If not, please add different numbers.

*In this figure only the hours were used in which data of both instruments was present. So AGM and EC pairs are used. The numbers in the top row indicate the number of pairs used to create the average.*

c.  4) Figure 11. there is lack of explanation on the diurnal pattern. There is a "jump" in the deposition flux in the morning at 5 am, was it really happening or just due to the noise? There are not many available data at night for deposition flux datasets (both only 2 at 5 am and 22 pm).

*We present this pattern as a sample result of our observations, as an example of which research area has now been opened by these observations. The exact explanation of the mechanisms behind these curves is a separate research project and not in the scope of this article. Yet, the overall behaviour is plausible, with moderate deposition during the night (humid soil, little convection) and stronger emission during daytime (dryer soil, higher temperature, stronger convection, lower concentrations).*

(22) Line 595. I would like to see how the footprint are associated with the surface roughness z0 and stability length L (stable and unstable conditions).

*We expect little effect of $z_0$ variability on the footprint over the Cabauw site during the campaign. Kljun's paper (https://gmd.copernicus.org/articles/8/3695/2015/) has discussed the sensitivity of the footprint for different values of $z_0$. During the campaign, NW & SW patches of grassland were slightly shorter than other patches due to the early June cut. We estimate canopy height differences among grass patches from 5 cm to 10 cm. Assuming $z_0$ is 13% of canopy height, $z_0$ difference range from 0.0065 m to 0.013 m, which is relatively small. Ditches were less than 10% of the land cover, with a roughness length of 0 m and evenly distributed among grasslands. In all, $z_0$ differences and changes were few and are expected to have little impact on the footprint.*

*Stability length L does impact the extent of the footprint: The footprint is larger in stable conditions (ca. 300 m) and smaller (ca. 200 m) in unstable conditions, see the plots below.*

[Figure]

*Figure 4. EC footprint within green and light-green wind sectors: (a) all data; (b) under stable conditions; (c) under unstable conditions. Note that the scale bar (50 m per unit) indicates that all figures share the same spatial scale.*

(23) Line 660 please indicate it in the Figure 1.

*Line 660 describes the DOAS container. This container is already indicated in figure 1, it is the centre of the graph.*

(24) Line 802. Please provide a footprint analysis for AGM, the EC measurement should be corrected with a footprint modelling.

*AGM footprint analysis: See our answer under (4)*

*Correction of EC flux measurement using footprint modelling: See also our reply under (1) and (2). Throughout this paper we have assumed the terrain relevant to our observations to be flat, with homogeneous vegetation, and the resulting flux field to be 3D-homogeneous. This is an approach that is commonly used in flux studies, but obviously it is an idealization that in practice does not hold completely. In this approach, footprint issues do not come into play.*

*During earlier campaigns for methane intercomparing $CH_4$ EC systems the Cabauw site proved to be really useful for that (Peltola et al., 2014 https://bg.copernicus.org/articles/11/3163/2014/) without the need for corrections using footprint modelling. However, specific for NH3 the terrain turned out to be more complicated than we anticipated.*

*The exact shape of the footprint and its orientation on the map is determined by meteorological conditions and can vary rapidly with time. Combining and unraveling the mix of the effect of a varying terrain with a varying footprint would be the ultimate challenge for the analysis of our measurements. Correction of the measured flux data with a footprint model is however only possible if all relevant data of each part of the footprint are available throughout the campaign period. This also includes proper knowledge of all agricultural activity in all subplots. We did however not collect these data because we did not anticipate we would need them. So for this campaign we cannot perform the correction requested by the reviewer. Yet, with all data available it would still be questionable if we would succeed in getting a better comparison between the data of AGM and EC. For this, the Cabauw site is possibly too complex and diverse.*

*In our current analysis footprint issues will result in somewhat different measurement results from the two instruments, and also in somewhat imperfect flux figures. But the results still show that the two techniques can provide half-hourly flux measurements that appear to be consistent. That is the main message of the paper.*

*We agree with the reviewer that further studies into the footprint issues would take us a step further, especially for the AGM (miniDOAS) measurement. The campaign has shown us the relevance. We therefore will address these issues again in a future campaign in the Ruisdael observatory "de Veenkampen". Here, the terrain will be much closer to the ideal homogeneous environment, and there will be no sources in the direct vicinity.*

**Technical comments**

(25) Line 410. Remove "respectively"

*Will do*

(26) Line 413. Add a comma, to be 5%, 5%, and 1%, respectively.

*Will do*

---

## Author Comment (AC2)

**Response to Reviewer 2**

**General**

We thank both reviewers for their thorough reviews and their wealth of suggestions. We greatly appreciate their input. We recognize in their reactions that the progress that we have made in the described research now opens up a large number of new questions: our research team felt exactly the same. With the reviewers help we now harvested a number of good ideas to further optimize the data evaluation of this experiment, but more important: of experiments to come. Because facing the editor's request to reduce the length of the paper substantially does not allow expanding the evaluation on all aspects highlighted by the reviewers. We have tried to find a balance between which parts of the review comments we take on board now, and which parts we will take with us to our next campaign.

A main decision is how to deal with the footprint issues raised when evaluating the data. In the past, the Cabauw site has worked very well as a location for many flux intercomparison experiments with other gases (e.g. Peltola et al., 2014 https://bg.copernicus.org/articles/11/3163/2014/). However, this campaign has shown that it is probably somewhat too inhomogeneous for detailed intercomparison of ammonia fluxes. This was aggravated by the wind directions during most of the campaign, which were different from the normally prevailing southwesterly winds. Considering that (1) we did not collect the detail of activity data in the direct surroundings for a proper evaluation of the impact of different footprints on the fluxes of EC & AGM, and (2) that we are urged by the editor to reduce the paper length considerably, we have decided to limit our analysis in this paper to the standard, 3D-homogeneous flux approach. We are aware that with this approach some of the observed differences will originate from footprint issues. These issues will get more emphasis and attention in our next campaign, that will be located in a more homogeneous area.

In the remainder of this document we will follow the reviewer's text, and our pointwise reactions and answers can be found in *italic blue text* at the applicable position.

On behalf of the author team,

Daan Swart.

Review of the paper „Measuring dry deposition of ..." by Daan Swart et al.

First I would openly state that I have a long lasting collaboration with some of the authors and I closely followed and discussed their progress. I cannot exclude that I am biased in this sense.

This paper presents an NH3 flux intercomparison with two novel open-path instruments. The major advantage of open-path instruments is the absence of an inlet line that is notoriously influencing high frequency concentration measurements needed to determine fluxes. The measurements took place at the Cabauw station located in the middle of the Netherland. It is an open landscape mostly used as intensive grassland. From a micrometeorological perspective it is good location as most of the time good turbulent conditions are present.

I am very pleased to see the great progress that was achieved regarding the accuracy and precision of the DOAS system. The systems have now reached a level that meaningful vertical gradient measurements are possible. Also the open path system from Healthy photons is precise and fast enough to perform reliable EC measurements.

Looking from a greater distance the overall results summarized with figure 11 are looking very nice. Both systems show similar NH3 fluxes and more important they show the same diurnal structure with identical changeover from deposition to emission. So it is very likely that both instruments do measure an existing mean flux over the surface determining the flux.

I could stop here and say publish as is with some minor corrections already addressed by the first reviewer. But this is not my understanding of a serious review and I will dig in the following a little bit deeper. But keep in mind that this is complaining at a high level.

Specific issues:

(1) I think that the title is not appropriate. The work focus on an intercomparison of two approaches to determine bidirectional NH3 exchange. The title should reflect this.

*We will adapt the title to:*
*Field comparison of two novel open-path instruments that measure dry deposition and emission of ammonia using flux-gradient and eddy covariance methods*

(2) It would be important to describe shortly the "flux landscape" of the chosen site. It is located in an area with intensive animal production associated with a high NH3 turnover. Generally small areas with high emission densities (stable, storage, manure fields) are surrounded with intensively managed field that are expected to show all the time changes from emissions to deposition mainly controlled by their actual N-status (see e.g. the year around measurements that we did at the Oensingen grassland site many years ago, Flechard et al., 2010 https://bg.copernicus.org/articles/7/537/2010/). Judging from a google maps picture from the area there are many small individual fields as well as stationary sources within the footprint area.

[Figure]

*We will move Figure S1 (below) to the main text and add a short description of the surrounding area at Cabauw to the section "Campaign setup and Site". This shows the complexity of the terrain. We received a similar request from the other reviewer.*

[Figure]

*Figure 1. The area surrounding the Cabauw measurement site. Cabauw is in a flat area at -1m, located in the delta of the river Lek shown in the southeast. The line with housing going east-west, running north of the tower has a series of farms. Map from [www.pdok.nl/](www.pdok.nl/) (downloaded 07-02-2021).*

(3) Deposition mainly occurs during night stable conditions when NH3 is concentrated in the boundary layer and emission occurs during instable conditions when the growing boundary layer leads to NH3 concentration below the "system compensation point". Consequently an inverse relation between concentration and fluxes must be expected, even though at a first glance this seems contra intuitive. In this context I was very puzzled by the sentence in line 488 *"The highest concentrations were observed by both systems at noon when air temperature reached the highest level of the day (Figure 8)".* I am uncertain whether this is just a slip of the pen or a misunderstanding.

*We thank the reviewer for noting this obvious mistake. Figure 8 indeed shows exactly the opposite, as he rightfully expects. We will change the corresponding line to: "The highest concentrations are observed during nighttime when the boundary layer height is small and vertical mixing is limited. During daytime the concentrations decrease due to the rise of the boundary layer and the increased vertical turbulent transport."*

(4) The evaluation of the EC data is state of the art. The most important correction is the high frequency damping in the order of 20 to 30%. I am aware that the empirical ogive method needs a lot of good data. I suggest to use only flux data with a good looking covariance function for both NH3 and temperature flux and not using daily mean values.

*As outlined in lines 354 to 357, half-hourly flux damping correction factors were filtered applying standard flux filtering criteria. Also, only half-hourly concentrations with OSS (HT light intensity) of less 40 % were filtered out. In doing so, the high-quality cospectra were used to estimate the damping. No further manual filtering of ogives and covariance functions was applied since that may lead to biased results.*

*Fluxes were corrected with their (own) half-hourly high-frequency correction factor if available as written in line 358. If the required half-hourly correction factor was missing, the daily median correction factor was used and not the average to avoid bias. No changes to the text will be made since the information is already given in the manuscript.*

(5) DOAS gradient measurements require a demanding procedure to be precise enough that includes measurements in cross position. I rate this as a very positive development of the DOAS approach that I was not expecting based on our own experiences. But I see a tendency to overestimate the precision of the DOAS measurements. E.g. in line 280 it is stated "*that differences can be measured well below our target precision of 0.1 µgm-3.*" In line 449 it is stated "*The random error of the miniDOAS NH3 concentration differences ... was determined to be 0.088 µgm-3*". This is below the target limit but not well below.

*We will adapt line 280 to correct this overclaim. The line will read: "In the Results section (Sect. 4.1), it will be illustrated that after these steps the pair was capable of measuring NH$_3$ differences within our target precision of 0.1 µg m$^{-3}$."*

(6) I also have an other interpretation of the cross position data.
   a. In the second period I judge that there is a systematic offset in the order of 0.14 µgm-3 as this is a consistent value for good wind conditions.

   *Over the full second cross period, we do not see a systematic offset, but rather an offset that varies gradually between 0 and -0.2 µgm$^{-3}$ during this period. Note that a similar meandering is seen in the wind direction. There is only a very limited number of points (n=5) in the green/light-green wind sector in this period. Similar 'outliers' also occur during cross periods 1 and 3, and may have a different cause, e.g. a very local source. Obviously, we cannot exclude a zero drift completely, but since the zero level in cross periods 1 and 3 is the same, and flux values directly after cross period 2 compare well with the HT-values, we trust the zero level of the miniDOAS instruments to be stable, even if we could not confirm this properly in the second cross period. The stability over many weeks was also confirmed by lab experiments prior to the campaign.*

b. In case the wind is coming from the red sector I don't see a mechanism that could produce the measured difference between the two crossed paths, especially for the lower concentration range. The obstacle upwind will affect mostly the turbulence and not the NH3 concentration.

*On the mechanism: the offset occurs when part of the path of the miniDOASses is affected by a turbulence plume, and another part is not affected or differently affected. For example: if the first half of the DOAS path is unaffected, and the second half of the path is completely well-mixed by the plume, about 25% of the existing gradient is observed instead of zero.*

c. You could also check the EC-NH3 covariance function and time series to see whether there is a major disturbance.

*We will check this and report on our findings later.*

(7) As far as I know there are many influences that potentially influences concentration determination in the order of 0.1 to 0.2 µgm-3 especially under field operation. Judged from figure S5 such a correction could at least partly explain the higher NH3 emission of the AGM approach around noon.

*It is clear that the determination of the flux by AGM is the most difficult at noontime when both concentrations and gradients are typically smaller due to the daytime convection and boundary layer rise. A small systematic offset in the zero of the difference measurement is more relevant at that time of day. When comparing figure 10 in the main text, and figure S7 from the supplement, it appears however that there is also an influence from the spreading of manure: results of the instruments are closer together at noon when no manure is spread. The issue is possibly also related to the interplay of an inhomogeneous source/sink field with the different footprints of the instruments. We aim to look into this further in our next campaign, which will be performed in a less complex terrain.*

(8) The whole footprint discussion should be modified. In case of the EC approach the calculated fluxes are representative for a small volume at a height of 2.8m. In case of the AGM approach it is a mean vertical flux integrated over a 22m path between 0.76 and 2.26m assuming that the used transfer velocities correctly reflect the atmospheric turbulence. These fluxes need to be translated into exchange fluxes at the soil surface. The simplest approach and implicitly used in most cases is that the vertical flux is constant in all 3 dimension and consequently the measured fluxes are equal to the surface exchange flux. In such conditions footprint considerations are not an issue.

*As already indicated under 'General' above, we agree with the reviewer that for now a relatively standard 3D-homogeneous approach is called for, and will adapt the manuscript accordingly where needed. We should however remain aware that the terrain of the campaign is not completely homogeneous, not in space and not in time, over the typical footprint of around 100 to 200 meter and the full campaign period. It is too patchy in surface parameters, sheep are present and their location is varied, grass is harvested from the land at some locations. This can result in differences in the measured fluxes of the two instruments. See also our answer to reviewer 1 under (24).*

(9) The footprint climatology shown in Figure 13 gives a good impression which area will on average determine the measured fluxes. But the analysis should be adapted to the selected data point (green and light green conditions).

*We agree and will include as Figure 13b the EC-footprint for only the points in the green and light-green sectors, as shown below.*

[Figure]

*Figure 2. EC footprint for (a) all wind sectors and (b) the green and light-green wind sectors*

(10) NH3 exchange fluxes will most likely be different between all the small fields in the neighborhood of the measuring point within the footprint. Consequently EC and AGM approach will considerably differ as the footprint density function for the two approaches differ. Note also that this function will considerably differ for the lower and upper DOAS path.

*We agree. A slightly more elaborate footprint evaluation was done using Kljun's online tool that uses satellite observations for land class identification (https://geography.swansea.ac.uk/nkljun/ffp/www/). That analysis already shows the differences in the land classification on a small scale around the tower. As stated already we lack activity data on the different plots within the footprint area, so we can state that this can cause differences but not more than that.*

*It is indeed also clear that the miniDOAS footprint of the upper and lower path will be different. Also, the fact of a line-average versus a point measurement plays a role in determining the footprint region of the AGM. We currently do not have a proper analysis for that but assume the AGM footprint will be slightly smaller than the EC footprint in our study because the miniDOAS instruments are on average lower than the EC. Obviously, all these issues disappear when we assume a 3D-homogeneous flux field in the analysis, as we do in this paper.*

(11) I also suggest to separately calculated the footprint density function for stable and instable conditions. By the way the bls-R tool developed by Christoph Häni is appropriate to do so. It allows to calculate concentration and flux footprints µgm-3 (https://github.com/ChHaeni/bLSmodelR). It would interesting to overlay these footprint density function on a land use map. This would allow a plausibility control whether the recorded differences in Figure 11 reflect a topographical effect or whether systematic

effects in the measurement systems are in the focus. But an in depth footprint analysis needs detailed information on the land use.

*We would definitely like to use this tool for upcoming experimental sessions where we will continue the evaluation of this campaign. As we need to shorten the paper at the editor's request we will however not include this exercise in this paper.*

To summarize I suggest to do:

- Reconsider the title
- Add a paragraph regarding the expected flux landscape at Cabauw. This can be integrated to the discussion of the concentration characteristics.
- Down scale the footprint issue. The presented fluxes assume a homogeneous vertical flux and that the used set of turbulence parameter reflect the present meteorological conditions. A consideration of the footprint is for another paper.

(12) For my curiosity I would like to see in a supplement or in the main paper a plot with the AGM concentration differences, the transfer velocities and the fluxes (extension of figure S5).

*Here is the figure.*

[Figure]

*Figure 3. The measured miniDOAS NH$_3$ concentration differences (µg m$^{-3}$), the transfer velocities (m s$^{-1}$) and the AGM NH$_3$ fluxes (µg m$^{-2}$ s$^{-1}$). In this graph, cross-periods are excluded and the semi-transparent lines show the data during u\*<0.1 conditions.*

(13) Regarding the EC data it would be helpful to see NH3'w' covariance function as well as ogives for a few cases, both from the green and red sectors. (not only the best!).

*Figure 3 shows ogives and covariance functions of the heat and NH3 flux from the yellow sector. The yellow sector was chosen since it can be treated as a compromise between good and bad flux conditions. No damping estimation was possible for (a) and (b) since their corresponding covariance functions (g) and (h) showed no distinct peak and a high variability at the sides making a reasonable flux estimation impossible. In example (c), damping with the ogive method would probably be underestimated since significant damping is given in the low-frequency range. Considering the time of the day, the issues in ogives and covariance functions were probably caused by fog or dew. Water droplets may have been accumulated on the mirrors. (d) to (f) and their corresponding covariances were measured around noon. They show the typical ogives observed at the site with the highest flux loss between 0.1 and 1 Hz. Figure 4 shows ogives and covariance functions of the heat and NH3 flux from the red sector. Compared to Fig. XX, flux damping during noon was lower. For cases (d) and (e), damping estimation is possible but associated with large uncertainties. No flux estimation was possible for example (l) of Fig. YY. Overall, damping estimation based on measured ogives/cospectra was possible for the different sectors. However, filters regarding time lag, flux quality flag, and instrument performance are mandatory to improve the quality of damping factors.*

[Figure]

*Figure 4. Flux ogives and covariance functions of the heat and NH3 flux of different half-hours from the yellow sector on the 13th of September. (a) to (c) and (g) to (i) refer to times from 8:00 until 9:30 UTC/LT, and (d) to (f) and (j) to (l) refer to times from 12:00 until 13:30 UTC/LT.*

[Figure]

*Figure 5. Flux ogives and covariance functions of the heat and NH3 flux of different half-hours from the red sector on the 8th of September. (a) to (c) and (g) to (i) refer to times from 11:30 until 13:30 UTC/LT, and (d) to (f) and (j) to (l) refer to times from 15:30 until 16:30 UTC/LT.*

---

## Author Response (AR2)

Dear editor,

We have submitted the revised version of our manuscript.

We are happy to receive your positive assessment of our manuscript. In your previous comments, you have given us very clear and concrete instructions on how we should further shorten the Discussion text. We'd like to thank you for that.

We have added the table to the Discussion you requested (Table 1), and shortened the Discussion text to 50 lines. Any text about instrument performance that we felt could be valuable for a portion of the readers we moved to the Supplementary Materials.

Furthermore, we have addressed the smaller comments about units and the inclusion of copyright information.

We look forward to hearing from you,

Kind regards,
On behalf of all authors,

Susanna and Shelley